# Sick leave or work sick? Examining the antecedents and conceptualizations of presenteeism and absenteeism among teleworkers during COVID-19: A scoping review

Behdin Nowrouzi-Kia[1,2,3,4]*, Sharada Nandan[1], Edris Formuli[1], Kishana Balakrishnar[1], Ali Bani-Fatemi[1], Aaron Howe[1], Yiyan Li[1], Luke A. Fiorini[5], Shane Avila[1], Chantal Atikian[1], Kathy Zhou[1], Mahika Jain[1], Basem Gohar[3,6]

1 Department of Occupational Science and Occupational Therapy, Temerty Faculty of Medicine, University of Toronto, Toronto, Ontario, Canada, 2 Krembil Research Institute-University Health Network, Toronto, Ontario, Canada, 3 Centre for Research in Occupational Safety and Health, Laurentian University, Sudbury, Ontario, Canada, 4 Institute for Mental Health Policy Research, Centre for Addiction and Mental Health, Toronto, Ontario, Canada, 5 Centre for Labour Studies, University of Malta, Msida, Malta, 6 Department of Population Medicine, University of Guelph, Guelph, Ontario, Canada

* behdin.nowrouzi.kia@utoronto.ca

## Abstract

Many organizations have shifted to hybrid or remote work arrangements in response to the COVID-19 pandemic. Illness, whether physical or psychological, can manifest during telework (remote or home-based work), leading to presenteeism and absenteeism behaviour. However, varying definitions of presenteeism and absenteeism have made measuring presenteeism, absenteeism, and their antecedents increasingly challenging. This scoping study seeks to define presenteeism and absenteeism in the (tele)workplace and systematically identify the factors contributing to their occurrence. A systematic literature search was performed on seven online databases: MEDLINE, CINAHL, PsycINFO, ABI Inform Global, SCOPUS, Web of Science and Business Source Premier. We applied the PRISMA-ScR guidelines and Joanna Briggs Institute framework to systematically collect, identify, and report studies. The inclusion criteria encompassed studies with participants aged 18 to 65 years old who currently work in a telework environment for at least 50% of their work hours. Of 826 initially identified studies, 18 studies were included after screening (11 quantitative, three qualitative and two mixed-methods studies). A total of 26,805 workers were included in this review across 16 empirical studies. Overall, presenteeism is defined as working while ill, and absenteeism is known as being absent from work or taking sick leave. We identified three major categories for the antecedents of presenteeism and absenteeism behaviour: organizational (i.e., job demand and telework), environmental (i.e., work and home environment), and individual (i.e., poor mental health and job perception). Presenteeism and absenteeism among teleworkers manifest from organizational, environmental, and individual forces that lead to working while sick, or being absent

**Data availability statement:** All the data used in this study are included within the article.

**Funding:** The author(s) received no specific funding for this work.

**Competing interests:** The authors have declared that no competing interests exist.

from work, respectively. We found that each of these antecedents relates to one another through the social determinants of health framework. Our conceptual findings guide developing top-down organizational policies and strategies that address presenteeism and absenteeism behaviour, particularly in telework settings.

## 1 . Background

The COVID-19 pandemic greatly impacted the way people work. From in-person to hybrid and fully virtual work arrangements, multiple sectors have embraced telework, such as the governmental, research, finance, IT, administrative, and office industries [1,2]. Telework is defined as working outside of conventional office settings (e.g., at home or a remote location) using information communication technologies (ICT) to communicate with others and conduct work-related tasks [3,4]. Telework has been used by organizations in the past, however with the emergence of the COVID-19 pandemic, there has been a surge in the utilization of remote work [5]. Many organizations opted to shift to remote work to maintain productivity while adhering to public health restrictions [2]. Before the height of the pandemic, around 4% of Canadians worked from home in 2016; however, those numbers have risen in January 2021 as 32% of Canadians aged 15–69 work most of their hours from home [6]. With public health restrictions being lifted, several organizations continued to practice remote work with about 20% of Canadians working from home in November 2023 [7]. As telework persists, it is crucial to explore its impact on workers.

COVID-19 has shifted organizational and individual attitudes toward working. Research has shown that the mandatory adoption of telework during the COVID-19 pandemic increased the likelihood of workers opting in for telework after the height of the COVID-19 pandemic restrictions, even among those initially hesitant about such mode of work (Baudot & Kelly, 2020), indicating a shift in preferences due to the pandemic. Regarding work-related productivity, some studies indicate a loss of productivity from telework [8,9], while others indicate no difference in work-related productivity when compared to in person work [10–12]. This paper examines productivity in telework during the COVID-19 pandemic and provides a clearer understanding of how it may have been impacted. Telework is associated with greater flexibility, reduced distractions from coworkers, and reduced travel time, making it a highly adaptable and preferred modality of work for most individuals [13]. Despite the benefits of telework, there have been some caveats that have manifested during COVID-19. Teleworkers have reported challenges such as technology issues (e.g., issues using new platform, technical issues), privacy concerns, equipment (e.g., difficulty obtaining the correct software and hardware to work remotely), engagement, and stress from working [14]. More broadly, teleworking has been associated with phenomena such as presenteeism and absenteeism in the workplace since the onset of COVID-19 [15].

Definitions of presenteeism and absenteeism are presented inconsistently in the literature, leading to different interpretations of the phenomenon [16]. Presenteeism can be understood as working while sick [17] or the pressure to remain online while

sick [1], and productivity loss due to working while sick [17,18]. Additionally, it has also been defined as being physically present but functionally absent [19,20], with Grigore [21] identifying stress from contracting COVID-19 as a reason for being mentally absent from but physically present at work. For absenteeism, definitions range from being physically absent from work [22,23], to lack of attention and productivity at work [24]. Moreover, voluntary absenteeism or "unauthorized absenteeism" is absence not related to illness, which is often seen as employees' avoidance of work [25]. It is important for researchers in this field to clearly define what constitutes presenteeism and absenteeism in work health contexts to provide a more accurate and standard definition of the phenomena that can be used in scientific studies.

Accurately capturing presenteeism and absenteeism is crucial for data to be used to understand behaviour and inform organizational policy related to productivity and occupational health interventions. This can be achieved by employing the social determinants of health model, a framework used in public health to guide top-down interventions that address population health and well-being [26]. Research conducted by Shafer et al [27] found that teleworking during the COVID-19 pandemic gave rise to presenteeism behaviours as it gave mildly ill workers the opportunity to work while adhering to public-health recommendations to stay home during illness. Concerns with absenteeism pre-dates the pandemic [28], but with the shift to telework during COVID-19, presenteeism became more prevalent as employees chose to work remotely while sick to avoid being absent from work and to reduce the transmission of diseases [3]. Both presenteeism and absenteeism behaviours have been found to reduce work-related productivity [17,19]. These phenomena are concerning for organizations as increased presenteeism can lead to increased transmission of infectious diseases and burnout, whereas increased absenteeism and presenteeism can result in non-infectious illness [18]. While health problems such as an illness are a prominent reason for presenteeism and absenteeism behaviours, psychological factors such as social support and coping mechanisms play an important role in determining whether an employee engages in such behaviours [20]. This scoping review aims to capture and investigate the various definitions of presenteeism and absenteeism, specifically focusing on their antecedents amongst teleworkers. The antecedents of presenteeism and absenteeism among workers who engage in teleworking have not been systematically identified, nor have the definitions of both phenomena. Our study will be among the first to systematically capture the factors that contribute to presenteeism and absenteeism among teleworkers in the workplace and present them in a theme-based narrative form. Creating a more concise definition of both terms may improve our understanding of presenteeism and highlight the potential factors why workers choose to engage in such behaviours. Understanding the current state of how researchers understand these work behaviours is beneficial as accurate measures of presenteeism and absenteeism can be developed within the context of telework to create robust methods that ease the transition to remote work post-pandemic. Doing so can allow for developing evidence-based, top-down interventions that target the antecedents that lead to presenteeism and absenteeism before such behaviours occur.

Identifying antecedents from the organizational level can be an effective strategy for early prediction of productivity losses and workers' ill health. Several studies have established that presenteeism and absenteeism are influenced by a wide range of complex factors [29–32]. Recognizing these broader influences, this study employs a social determinants of health (SDH) framework to explore solutions that address both the underlying and systemic factors contributing to these workforce challenges. By adopting this approach, we aim to develop interventions that go beyond illness management and encompass workplace policies and socio-economic conditions that impact workforce retention and overall well-being. As such, our study seeks to investigate the following: 1) How do we define presenteeism and absenteeism among adult teleworkers? 2) What are the antecedents of presenteeism and absenteeism among adult teleworkers?

## 2 . Methods

### 2.1 Study design

Our scoping review followed the methodological framework proposed by Arksey and O'Malley [26]. The purpose of a scoping review is to collect and map key concepts in the literature within a particular field of interest to address a broad

research question [26]. We employed a descriptive-analytical approach, providing a numerical summary and categorizing findings to theorize and contextualize results. By exploring definitions of presenteeism and absenteeism in the context of telework and their antecedents, a systematic literature search was conducted following the reporting guidelines outlined in the Preferred Reporting Items for Systematic Reviews and Meta-Analyses extension for Scoping Reviews (PRISMA-ScR) framework [33]. Before commencing, the scoping review protocol was registered with the Open Science Framework in June 2023 (Registration DOI: https://doi.org/10.17605/OSF.IO/UR5A6).

## 2.2 Search strategy

Electronic databases including MEDLINE, CINAHL, PsycINFO, ABI Inform Global, SCOPUS, Web of Science, and Business Source Premier were searched to identify empirical studies related to our research objectives. Key search terms included teleworking (e.g., "telecommuting" and "remote work"), presenteeism (e.g., "sickness presenteeism") absenteeism, definition, and COVID-19. Refer to the full search history in S1 Data. The search was conducted by S.A and C.A on 02-10-2023 and concluded on 14-12-2023. A University of Toronto Health Science librarian was consulted for the selection of appropriate databases and to ensure comprehensiveness and appropriateness of each search term to mitigate selection bias.

## 2.3 Eligibility criteria

The study's inclusion and exclusion criteria were simplified in Table 1. The specific timeframe was chosen to capture studies that examined presenteeism and absenteeism in the context of COVID-19. Studies which met the following criteria were included: 1) adults aged 18–65 years old; 2) currently employed individuals; 3) studies that examined telework during the COVID-19 pandemic (2020 and onwards); 4) studies published in English; 5) studies that discussed antecedents of presenteeism and/or absenteeism during telework; 6) studies that discuss definitions of presenteeism and/or absenteeism; 7) empirical studies (experimental, observational, secondary, qualitative, mixed methods) and 8) employees working in telework for at least 50% of working hours will be considered. Telework is operationalized as working from a place outside of the office, using electronic devices such as a PC or a laptop to communicate with fellow workers. The exclusion criteria are as follows: 1) non-working populations (children and older adults who are retired); 2) tertiary studies (e.g., literature reviews); 3) non-empirical studies; 4) non-peer reviewed journal articles; and 5) studies that moved to telework prior to the COVID-19 pandemic due to technological and workplace differences [34]. Further, we excluded studies examining in-person work environments and a move to telework arrangements before 2020. Since telework arose in prominence and popularity across workplaces from the 1990s onwards [34], we wanted to ensure that organizations that implemented telework did so due to COVID-19. While the literature indicates that individuals aged 15–69 work remotely in Canada, the age range was limited to 18–65 as many countries considered individuals below 18 as youths, often working

**Table 1. Inclusion and exclusion criteria.**

| Study characteristic | Inclusion criteria | Exclusion criteria |
|---|---|---|
| Population | Employed individuals<br>Aged 18–65 years of age<br>Working in telework for at least 50% of working hours | Non-working age populations (e.g., older adults or children) |
| Study Framework | Empirical studies (quantitative or qualitative using either primary or secondary data)<br>Discuss definition of presenteeism and/or absenteeism<br>Discuss antecedents of presenteeism and/or absenteeism | Tertiary studies (e.g., knowledge syntheses)<br>Non-empirical studies<br>Non-peer reviewed studies |
| Timeframe | Studies that examined telework during the COVID-19 pandemic (2020–2023) | Studies that moved to telework prior to the COVID-19 pandemic (e.g., in-person work and/or before 2020). |

PLOS Mental Health

occasionally, however, we aimed to study adult workers involved in telework during the pandemic [European Agency for Health and Safety, 2023]. Additionally, the exclusion of non-English articles and studies from non-Western regions may limit the generalizability of the findings.

We utilized the Population-Concept-Context (PCC) framework to guide the development of our research question [35]. The PCC highlighted important principles needed for the literature search such as population characteristics (e.g., participant age, demographics), the concept or focus of the scoping review, and context of the scoping review (e.g., the location, timeframe of the studies) [35]. The study population was individuals aged 18–65 working in telework for at least 50% of their working hours. The first concept was definitions of presenteeism, and absenteeism used in the study. The second concept was identifying the antecedents of presenteeism and absenteeism. Finally, the context was telework during the COVID-19 pandemic. We included studies published from January 2020 to December 2023 (inclusive) to fully capture the impacts of the COVID-19 pandemic. This timeframe helped us study teleworkers in the context of the pandemic, while excluding studies done on teleworkers before COVID-19.

## 2.4  Study selection

Our study adheres to the guidelines outlined by the PRISMA-ScR (Preferred Reporting Items for Systematic Reviews and Meta Analyses – Scoping Review) criteria [33] and the scoping review methodological framework proposed by Arksey and O'Malley [26]. Potential studies were organized using the reference manager software Zotero version 7.0 [36], and all studies were entered into the screening software Covidence [37]. Following removing duplicate studies, three reviewers (S.A, K.Z, S.N) independently conducted phase 1 title and abstract screening and phase 2 full-text screening. The inclusion and exclusion criteria were followed to guide the screening process, in which reviewers had regular meetings to discuss questions regarding the study selection. Discrepancies were resolved using a team-based consensus approach. The team was supervised and met regularly to discuss with an occupational health expert (B.N.K) to ensure consistency in the inclusion and exclusion of studies. The full screening process was then reported using a PRISMA flow diagram [33], refer to Fig 1.

## 2.5  Data synthesis

Data charting and reporting were performed systematically using the Arksey and O'Malley Framework [26]. Data was collected and sorted by involving obtaining key information from the selected research articles and sorting material into pre-developed categories [26]. A data charting table was created through an iterative process, where changes were made to the template as discussions surrounding the information desired to be extracted from articles occurred. The data charting table was piloted before implementation, where team members conducted pilot screening on studies. The data charting table included study details such as: study name, name of authors, year of publication, country of origin, study sample and study population, antecedents for presenteeism/absenteeism, the definition of presenteeism and absenteeism used within the study, and common themes identified. The results section below summarizes the extraction table (Table 2).

Category-based analysis was done using an analytical method to provide a narrative account of the studies. Three reviewers independently extracted data from each study to minimize bias. After initial data charting, the reviewers cross-checked each chart to ensure the results and categories identified were consistent between reviewers. Two independent senior reviewers (B.N.K. & Y.L.) resolved conflicts during team meetings to ensure consistency between reviewers. The definitions of presenteeism and absenteeism were taken directly from studies without further interpretation for the results section, and gaps were identified in definitions in the discussion section.

An iterative process was applied when devising the categorization of topics. This involved regular team meetings to share the categorization process. Categories and sub-categories were formulated based on common patterns examined during the data charting process, such as a pattern of multiple studies discussing factors surrounding poor mental health,

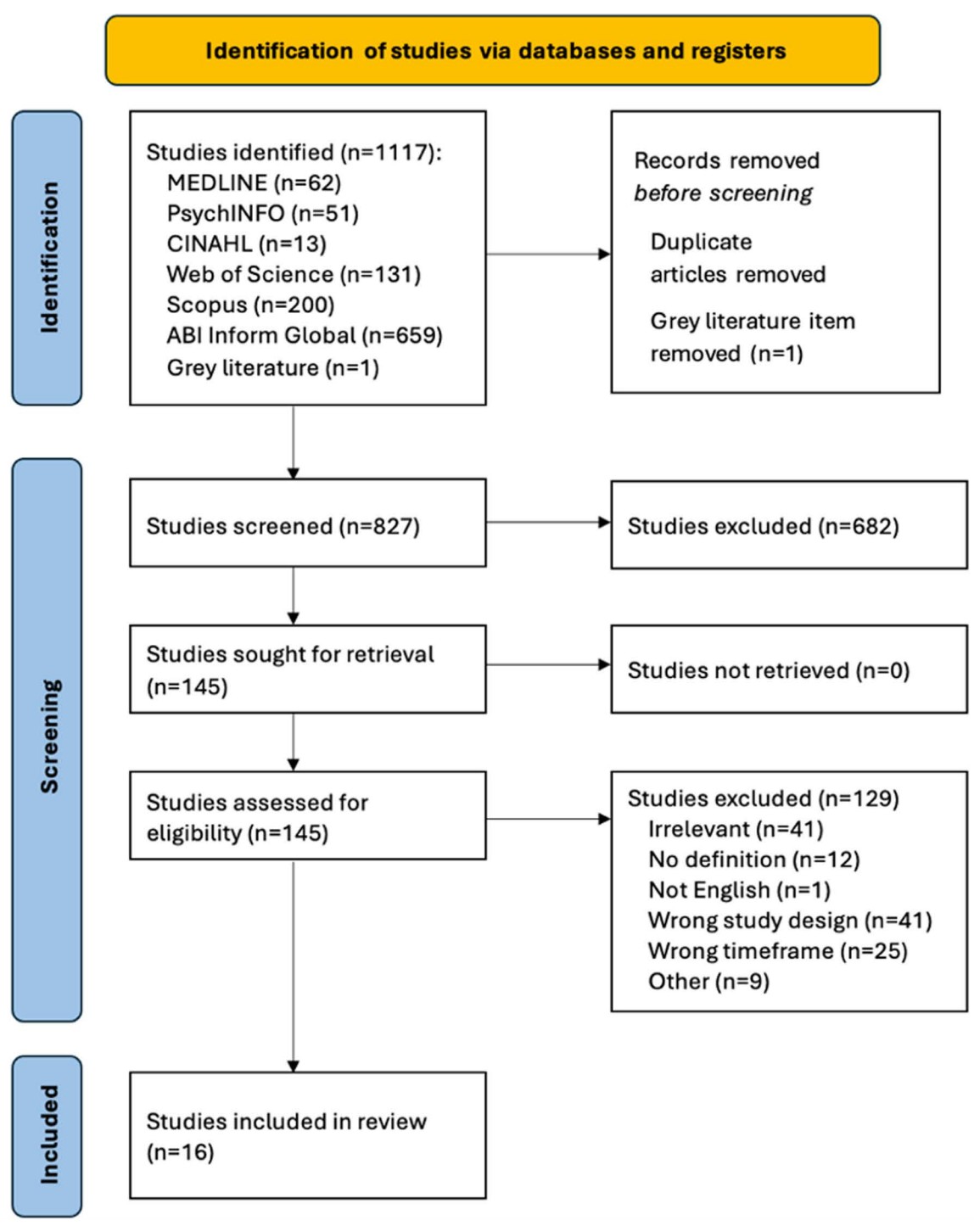

**Fig 1. PRISMA flow chart outlining the screening process.**

**Table 2. Summary of the studies included per the eligibility criteria (n = 16).**

| Author | Year | Country | Study Design | Target Population | Sample Size | % Women | Study findings |
|---|---|---|---|---|---|---|---|
| Adisa et al [1] | 2023 | United Kingdom | Qualitative (Semi-structured interviews) | Employees (i.e., higher education, accounting and finance, sales, marketing, and management employees), working from home during the pandemic | n = 32 | 53 | Presenteeism Definition: Employees are pressured to always be available online and respond to work-related tasks.<br>Reasons for Presenteeism: Stress of getting fired; showing dedication and worth to employer; constant access to internet; mistrust; work-life imbalance; poor online interaction with colleagues; increased job demands. |
| Biron et al [2] | 2021 | Canada | Quantitative (Cohort) | Telework employees from construction, manufacturing, service, health/ social aid, education, governmental, finances/insurance sectors | n = 275 | 57.4 | Presenteeism Definition: Working while being ill.<br>Absenteeism Definition: Work hours missed because of health issues (physical or emotional); leaving early because of health issues.<br>Reasons for Presenteeism: Negative psychological workplace environment; time constraints; quantity of work; excessive job demands.<br>Reasons for Absenteeism: Health problems. |
| Borge et al [38] | 2023 | Norway | Quantitative (Cross-sectional) | Day-time workers who performed their task in an office | n = 4,329 | 46 | Presenteeism Definition: Working while ill.<br>Absenteeism Definition: Sickness absence where employees self-certify four times a year for up to three consecutive days.<br>Reasons for Absenteeism include office design (shared vs. non-shared rooms) and lack of access to telework or working from home accommodation. |
| Brosi and Gerpott [39] | 2023 | United Kingdom | Quantitative (Experimental) | Full-time employees who worked in office or from home | $n_1$ = 138<br>$n_2$ = 274<br>$n_3$ = 242 | $n_1$ = 58<br>$n_2$ = 57<br>$n_3$ = 51 | Presenteeism Definition: Working while having an illness that warrants sickness absence<br>Reasons for Presenteeism: Guilt towards colleagues; guilt towards own health; workload; affective commitment. |
| Gerich [40] | 2022 | Austria | Quantitative (Cross-sectional) | Employees working in three economic sectors (finance, insurance, and IT) with access to telework | n = 886 | 46.5 | Presenteeism Definition: Working despite having an illness that justifies sick leave.<br>Reasons for Presenteeism: Work intensification and goal attainment in the workplace. |
| Grigore et al [21] | 2020 | Romania | Qualitative | Employees working in IT tech companies | n = 233 | 59 | Presenteeism Definition: Mental, rather than physical, absence from work.<br>Absenteeism Definition: Lack of physical presence in the workplace.<br>Reasons for Presenteeism: Stress from unhealthy working environment; working hours interfere with family dynamics; COVID-19-related stress.<br>Reasons for Absenteeism: Job dissatisfaction; unsupportive work environment; lack of recognition; inflexibility with procedures; stress from an unhealthy work environment. |
| Keightley et al [41] | 2023 | United Kingdom | Qualitative (Semi-structured interviews) | UK adults aged 18 and over working at home full time during COVID-19 pandemic | n = 27 | 70 | Definition of Presenteeism: Participants feeling compelled to remain digitally present.<br>Reasons for Presenteeism: Pressure/fear from colleagues to show that they are not "slacking"; excessive workload. |
| Mauricio and Laranjeira [42] | 2023 | Portugal | Quantitative (Cross-sectional) | Full time employee working at a nonprofit institution *Private Social Solidarity Institution* | n = 71 | 95.8 | Presenteeism Definition: Working while ill.<br>Absenteeism Definition: Absent during work due to a health problem.<br>Reasons for Presenteeism: Greater flexibility; wanting to appear productive; desire to complete tasks.<br>Reasons for Absenteeism: Physical health (musculoskeletal injuries, accidents at work, etc.) and mental health (depression, anxiety, burnout) problems. |

*(Continued)*

**Table 2.** (Continued)

| Author | Year | Country | Study Design | Target Population | Sample Size | % Women | Study findings |
|---|---|---|---|---|---|---|---|
| Michael [43] | 2021 | United States | Quantitative (Cross-sectional) | Full time or part time employee in the US for at least 1 year age 18 or older, and work telework for at least 8 hours a week for the past 6 months. | n = 217 | 62.7 | Presenteeism Definition: Going to work physically, however being absent mentally due to illness. |
| Okawara et al [44] | 2023 | Japan | Quantitative (Cohort) | Japanese workers who worked from home | n = 2,530 | 36.7 | Reasons for Presenteeism: lack of room/space for concentration; lack of light and foot space; inadequate temperature and humidity; use of a sitting table. |
| Ruhle and Schmoll [17] | 2021 | Germany | Quantitative Study | German working population including university employees and trade union members | n = 505 | 65.1 | Presenteeism Definition: Working in a state of ill-health. Reasons for Presenteeism: Constraints of absenteeism; job demands. |
| Ryoo et al [45] | 2023 | South Korea | Quantitative (Cross-sectional) | White-collar wage employees in Korea | n = 12,354 | N/A | Presenteeism Definition: Working while ill. Absenteeism Definition: Absent from work due to health problems. Reasons for Presenteeism: Flexibility of the working from home arrangement. |
| Shafer et al [27] | 2023 | United States | Quantitative study (Cross-sectional) | Adults 18–69 years of age seeking testing at COVID-19 testing sites or ambulatory medical care) for ARI (<10 days' duration) manifesting as fever, cough, or loss of taste or smell. | n = 947 | 72 | Presenteeism Definition: People showing up to work despite having an illness that requires rest and absence from work. Reasons for Presenteeism: Telework experience. |
| Takayama et al [46] | 2023 | Japan | Quantitative (Cohort study) | Working-aged desk workers (18–57) who never experienced working from home before the COVID-19 | n = 3,532 | 42.3 | Presenteeism Definition: Feeling unhealthy and going to work. Reasons for Presenteeism: Flexibility |
| Walker et al [47] | 2023 | United Kingdom | Mixed Methods | Office workers, at least 18 years of age, and working from home at least some of the time. | n = 140 | N/A | Presenteeism Definition: Reduced job performance due to illness. Reason for Presenteeism: Guilt |
| Yildirim [48] | 2022 | Global (France, Italy, Turkey) | Mixed Methods | Employees teleworking for the first time due to COVID-19 | n = 73 | 56 | Presenteeism Definition: Loss of productivity. Reason for Presenteeism: Isolation; concentration issues; boredom; heavy workload. |

which was categorized into a broader individual factors category. The original findings were categorised into five categories: COVID-19-related, epidemiologic, individual, environmental, and organizational factors. For example, if a study discussed attitudes on working from home and job perceptions, findings would be categorized under "individual factors". After discussion with the research team, it was decided to merge the COVID-19 and epidemiologic categories with individual factors, which includes individual, behavioural, and psychological components to health and wellbeing. The research team drew on internal expertise for this, in particular that of B.N.K. and that of Y.L. (an experienced social health researcher).

Numerical analyses using descriptive statistics were computed to present geographies, sample sizes, and proportion of females across studies. Simple numerical counts and percentages were utilized to explain the studies' geographic settings, methodologies, sample sizes, and gender breakdowns. Total sample size was conducted by summation of sample sizes across included studies. Geographic prevalence of studies investigating presenteeism and absenteeism was calculated.

## 3 . Results

### 3.1 Descriptive findings

After database searches, 1,117 studies were identified, from which 289 duplicates were removed. One additional study was identified from grey literature, which was also removed, resulting in 827 studies for title and abstract screening. Following the initial screening, 682 studies were excluded as they did not meet the inclusion criteria. The remaining 145 studies satisfied the inclusion criteria and underwent full-text screening. Ultimately, 16 studies were included in this review [1,2,17,21,27,38–48]. It is noteworthy that eight studies were removed after data extraction upon further investigation: one study was a tertiary source, four studies lacked evidence of antecedents of presenteeism and/or absenteeism, as well as a definition of either phenomenon, and three studies fell outside of the timeframe. Fig 1 displays the PRISMA chart depicting the full screening process.

### 3.2 Study characteristics

Majority of the studies (n = 13) were from European and North American continents [1,2,17,21,27,38–43,47,48]. One study had a global focus [48] with the other 15 having a single-country focus [1,2,17,21,27,38–47]. Amongst the latter, the most represented countries were the United Kingdom (n = 4) [1,39,41,47] followed by Japan (n = 2) [44,46] and the United States (n = 2) [27,43]. The remaining countries, each represented by a single study, were Austria [40], Canada [2], Norway [38], Portugal [42], Romania [21], Germany (n = 1) [17] and South Korea [45]. 'Global' refers to studies that had respondents from multiple countries, where authors used a global survey to understand presenteeism. Study characteristics of the included studies are summarized in Table 2.

From a methodological point of view, most of the studies (n = 11; 69%) were quantitative [2,17,27,38–40,42–46] with six (38%) of those being cross-sectional [27,38,40,42,43,45], three (19%) being cohort studies [2,44,46], one (6%) each being, respectively, an experimental study [39], and a study which did not specify what precise method was used [17]. Three (19%) of the studies were qualitative [1,21,41], with two (13%) of these involving semi-structured interviews and [1,41] one (6%) not specifying the precise method used [21]. The remaining two studies (13%) were mixed-method studies. All of the studies (16; 100%) provided gender breakdowns [1,2,17,21,27,38–48], with female representation ranging from 26% to 96% with a median value of 57%. Age was not reported in Table 2 as this study's inclusion criteria specifies working adults aged 18–65. Data from the studies were categorized based on their definitions of presenteeism, absenteeism, and their reasons for presenteeism and absenteeism.

### 3.3 Definitions of presenteeism and absenteeism across studies

**3.3.1 Presenteeism.** Thirteen of the studies defined presenteeism [1,2,17,21,27,38–40,43,45–48]. Across most studies, the definition of presenteeism remained consistent, with most defining it as "working while ill" [17,27,38–40,42,43,46]. Moreover, two studies emphasized that illness should warrant sick leave [39,40]. See Table 3 for a summary of the themes.

Similarly, a study defined presenteeism as being physically present in work but being mentally or emotionally absent due to sickness, stress, injury, or family obligations [21]. This understanding of presenteeism is more flexible, considering social obligations, psychological state, and working while sick. Essentially, these definitions of presenteeism consider the state of an individual's health while working.

Two studies reported another manifestation of the definition known as online presenteeism, which can be defined as "a situation where employees feel under pressure to always be available online and responding to work-related tasks" or "[an] employee is online or logged into the working platform but does other things, like taking care of a toddler or an elderly person" [1,21]. These two definitions contrast from the other studies as they do not involve employees working while ill, showing how differently presenteeism can manifest during telework. One study, which explicitly did not define the

**Table 3. Summary of major themes and sub-themes for presenteeism.**

| Themes | Sub-themes | Studies |
|---|---|---|
| Organizational antecedents | Excess job demand (n=5) | Adisa et al [1]; Ruhle and Schmoll [17]; Gerich [40]; Mauricio and Laranjeira [42]; Yildirim [48] |
| | Telework (n=5) | Ruhle and Schmoll [17]; Shafer et al [27]; Gerich [40]; Ryoo et al [45]; Takayama [46] |
| Environmental antecedents | Work Environment (n=3) | Adisa et al [1]; Grigore [21]; Yildirim [48] |
| | Home Environments (n=2) | Mauricio and Laranjeira [42]; Okawara et al [44] |
| Psychological/Personal/Behavioral antecedents | Poor Mental Health (n=1) | Biron et al [2] |
| | Stress (n=2) | Adisa et al [1]; Mauricio and Laranjeira [42] |
| | Anxiety/Worry (n=1) | Adisa et al [1] |
| | Fear (n=4) | Adisa et al [1]; Grigore [21]; Keightley et al [41]; Mauricio and Laranjeira [42] |
| | Guilt (n=2) | Brosi and Gerpott [39]; Walker et al [47] |
| | Time Management Issues (n=2) | Adisa et al [1]; Yildirim [48] |
| | Goal-directed behavior (n=2) | Gerich [40]; Mauricio and Laranjeira [42] |
| | Perception of Productivity (n=3) | Mauricio and Laranjeira [42]; Keightley et al [41]; Walker et al [47] |
| | COVID-related Stressors (n=1) | Mauricio and Laranjeira [42] |
| | Job dissatisfaction (n=1) | Mauricio and Laranjeira [42] |
| | Affective Commitment (n=2) | Brosi and Gerpott [39]; Mauricio and Laranjeira [42] |

term, understood presenteeism as solely "a loss of productivity" [48], while another study measured presenteeism as a general assessment of an individual's productivity and how health problems interfered with it [47].

**3.3.2 Absenteeism.** Four of the studies reported a definition for absenteeism [2,21,38,42]. The understanding of absenteeism is as follows: taking sick leave from work due to illness [38,42]; being late or leaving early because of physical or emotional illness [2]; being absent from the workplace [21]. Biron et al [2] did not give an explicit definition of presenteeism but measured absenteeism using a survey question: "*number of work hours missed because of a physical or mental health issue being late or leaving early because of health issue (past 7 days)*". Illness was understood as a physical or mental health issue. Borge et al [38] understood absenteeism as sickness absence from work, where employees do not engage in work activity due to sickness. Employees followed the Norwegian standard definition of sick leave: "*The general rules for sickness absence permit employees to self-certify four times each year for up to three consecutive days, while the [Inclusive Workplace Agreement] permits employees to self-certify 24 days in total during a 12-month period, where each spell can last up to 8 days*" [38]. Grigore [21] used the following definition "Lack of presence from workplace even though there is a social expectation for the employees to be there" which includes conceptions of the social norm of being at the workplace. Mauricio and Laranjeira [42] measured absenteeism as sick leave, being absent from work that day because of a health problem. The following studies defined illness/sickness as follows: being physically or emotionally unwell [2], sickness in general, undefined [38,42].

### 3.4 . Antecedents for presenteeism and absenteeism

Based on the data presented in the articles, we identified three types of antecedents, namely: (1) organizational; (2) environmental; and (3) individual. Organizational antecedents included factors such as policies, job demands, and workplace compensation [49]. Please refer to Table 4 for a summary of the absenteeism themes. Environmental antecedents refer to the environmental factors (i.e., physical conditions in a built environment) that contribute to why an event occurs [50].

**Table 4. Summary of major themes and sub-themes for absenteeism.**

| Themes | Sub-themes | Studies |
| --- | --- | --- |
| Organizational Antecedents | Limited access to telework (n = 1) | Borge et al [38] |
| Environmental Antecedents | Work environment (n = 2) | Borge et al [38]; Grigore [21] |
| Psychological/Personal/Behavioral antecedents | Stress (n = 1)<br>Poor mental health (2) | Grigore [21]; Biron et al [2]; Mauricio and Laranjeira [42] |
| | Job dissatisfaction (n = 1)<br>COVID-related stressors (n = 2) | Grigore [21]; Grigore [21]; Shafer et al [27] |

Throughout the literature, we identified multiple environmental factors that contribute to presenteeism and absenteeism during telework, which can be divided into work environment and home environment. We define work environment as the settings and conditions in which you perform job duties [51], whereas home environment refers to a living situation (i.e., working from home). Individual level antecedents, as defined from the social determinants of health model [52], include individual behaviours, mental health, and general health and wellbeing of the individual. These individual behaviours can be influenced by organizational antecedents as it impacts how employees respond to workload pressures.

**3.4.1 Organizational antecedents (job demand, telework).** Nine studies found that organizational antecedents can be attributed to high presenteeism and absenteeism rates among teleworkers [1,17,27,38,40,42,45,46,48]. Organizational antecedents for presenteeism behaviours included excess job demands [1,17,40,42,48] and the structure of telework. While telework gave workers the opportunity to work in a more flexible manner and reduce the risk of infection, it also added pressure on them to work while even moderately ill. [17,27,45,46]. Meanwhile, reduced access to telework was an organizational antecedent for absenteeism among teleworkers [38].

Six of the studies found that, due to high job demands, workers felt compelled to continue working despite illness [1,17,40–42,48]. With the COVID-19 pandemic and public health restrictions in place, many employees were laid off which increased pressure for the remaining employees with workers reporting a significant increase in telephone calls, emails, online meetings, and training [1]. Keightley et al [41] reported that participants, whilst working remotely, would be faced with several meetings and calls during their worktime amidst the pandemic which prevented them from completing any required tasks. Not to mention, those meetings also assigned them additional tasks that they were expected to complete [41]. As a result, ill workers who were unable to complete tasks would engage in presenteeism behaviours to avoid an increasing workload [1,42].

The way telework is structured and organized for workers creates less barriers to work despite feeling unwell, thus increasing the likelihood of them engaging in presenteeism behaviours [17,27,40,45,46]. Telework offers greater flexibility, where workers can adjust their work schedules and breaks to suit their health condition and other personal needs [17,45]. For instance, teleworkers were able to take breaks and rest in bed when sick [17,45]. A study by Gerich [40] found that management strategies such as indirect work control and telework led to presenteeism. Indirect work control (goal-directed management) encouraged employees to increase work efforts to ensure goal-attainment [40]. Due to goal attainment, increased effort was seen through the use of telework as employees intensified their work through working overtime and working despite sickness while working from home [40]. One study by Borge et al [38] assessed the relationship between telework access and sickness absenteeism and found that workers who had lower access to telework showed a higher likelihood of absenteeism.

**3.4.2 Environmental antecedents (work environments, home environments).** Five studies found that environmental antecedents contributed to presenteeism behaviours among teleworkers [1,21,42,44,48], with two studies identifying environmental antecedents to absenteeism [21,38].

In the context of presenteeism, three studies addressed environmental antecedents in relation to work environments [1,21,48]. Adisa et al [1] reported that employees working from home had a harder time adapting to their new workspace. Most employees struggled to convert their homes into a dedicated space for work or did not have a manageable space to balance both familial obligations and work-related activities. Being unable to adapt to new working environments is associated with presenteeism behaviour as many individuals found it hard to maintain a proper work-life balance as it negatively impact their performance and engagement [1]. Additionally, due to the nature of working remotely from home, many individuals reported that they invested more time and resources into their job since they were managing family-related activities (e.g., home schooling and childcare responsibilities) and coping with new work environments [1]. Another study by Grigore [21] found that stress from an unhealthy work environment led to presenteeism, with working hours and duties intersecting with personal and/or family relations outside of work. Many participants reported how isolation due to working from home was associated with concentration issues, distractions, and boredom which were associated with presenteeism behaviours such as being present but disengaged during work [48]. Ultimately, productivity decreased when workers were away from a physical work environment, which contributed to presenteeism behaviours [48].

In terms of absenteeism behaviours, there were two studies that reported environmental antecedents in relation to work environments [21,38]. According to Borge et al [38] who examined employees that either teleworked or performed their work from the office, office design impacted absenteeism behaviours. They discovered that those who worked in a conventional open plan office (i.e., shared office space) were more likely to engage in absenteeism compared to employees working in a private office [38]. Within the organizational environment, Grigore [21] found that employers often fail to recognize causes of absenteeism (e.g., depression and anxiety), and often underestimate the impact these factors have on job attendance and meeting job responsibilities. They also found that when the work environment does not encourage relationships between colleagues and work is unsupportive, it can result in absenteeism [21].

Two studies reported environmental antecedents of presenteeism in relation to home environments [42,44]. Home environments include an individual's living situation (i.e., number of occupants in a household) and the state of one's environment when working from home. According to Mauricio and Laranjeira [42], factors such as living with a partner contributed to presenteeism behaviours in teleworkers from Portugal. Moreover, a lack of room/space to work, inadequate lighting and ventilation, uncomfortable temperature, and prolonged sitting at a table were reported to cause work function impairments leading to presenteeism while working from home [44].

**3.4.3  Individual antecedents (mental health, guilt, job perception, COVID-related).**  The studies reported antecedents for presenteeism behaviour including poor mental health [2], guilt [39,47], time management issues [1,48], lack of goal directed behaviour [40,42], work productivity [41,42,47], COVID-related stressors [42], and job perception (e.g., job dissatisfaction) [42]. Antecedents for absenteeism behaviour include poor mental health [2,42], job perception (e.g., job dissatisfaction and insecurity) [21] and COVID-related stressors [21,27].

One study identified generally poor mental health of workers as an explanation for presenteeism behaviour [2]. Three studies identified stress related to job insecurity as antecedents to presenteeism, whereby job insecurity creates psychological distress and the pressure to continue to work virtually while ill due to fear of unemployment [1,42]. Anxiety and worry are an antecedent for presenteeism behaviour, specifically online presenteeism where an employee feels compelled to constantly be online [1]. Four studies identified fear as an antecedent [1,21,41,42], with one study emphasizing fear of unemployment as a driver for engaging in presenteeism, working while sick [42] and one highlighting fear that pressures people to always stay present on the job [41]. These factors contributed to the occurrence of presenteeism behaviour as they are associated with the perceived need to be constantly available at work virtually, in line with online presenteeism. A qualitative study found that the sudden transition from in person to telework modes of working during the pandemic creates stress [1]. Engaging in telework depletes social and personal resources that would have otherwise been obtained through in person work, creating poor employee engagement as a component of presenteeism behaviour [1].

Two studies identified general poor mental health as a reason for absenteeism [2,42]. Mauricio and Laranjeira [42] identified that around 8% of participants reported that their sick leave absence from work was due to mental health reasons, such as depression, anxiety, or burnout. Biron et al [2] revealed "health problems" as a reason for absenteeism, where health problems were understood as "any physical or emotional problem or symptom". Both studies understand absenteeism as absence due to either physical or mental health reasons [2,42].

Guilt was another contributing factor for presenteeism [39,47]. Brosi and Gerpott [39] noted that guilt was associated with presenteeism as they felt that their own action of not showing up forwork would fail the organization standard. Employees describe guilt towards colleagues as feeling that they have let their colleagues down if they choose not to work from home while ill. Overall, employees choose to work from home despite being ill to avoid the guilt they may experience if absence is taken. Walker et al. [47] noted that employees feel guilt when they do not appear to be online, which leads to reduced rest breaks and reduced productivity.

Poor time management is an antecedent of presenteeism, as employees may feel stress when they feel that they do not have enough time to complete tasks of heavy workload and start worrying about time [1,45]. An increase in the assigned number of tasks increases time stress, and performance in completing the work decreases consequently [48]. Yildirim [48] further explored presenteeism arising from working from home noting concentration issues, boredom, distractions at home, technical difficulties as factors that decrease work productivity. Adisa et al. [1] also focused on online presenteeism, working from home, and noted the difficulty in balancing work and life, and a general poor adaptation to work from home environments. Employees found it challenging to maintain good routines while working from home [1]. Mauricio and Laranjeira [42] revealed a strong relationship between job dissatisfaction and presenteeism; employees arriving at work with physical or psychological illness would perform work tasks at a lower ability, poorer attention and a lack of involvement. In addition, employees indicated a high tolerance of poor working conditions and a "lack of possibilities" of other opportunities as contributors to being dissatisfied with one's job and could relate to poorer productivity in the workplace [42]. Grigore [21] stated that the more satisfied an employee is with the job, the less absence there is. Job dissatisfaction sources that lead to absenteeism include unsupportive supervisors, not being acknowledged for work or skills in the workplace, and inflexibility with procedures used to accomplish work, such as solely in person modalities of completing tasks that could otherwise be completed at home. Grigore [21] further stated that perceiving the workplace as unsupportive and unfriendly can lead to absenteeism.

Adisa et al [1] found that an increase in job demands due to the pandemic was reported by participants, possibly due to the shift from in person to online work where employers expect employees to complete more work in a limited amount of time. Grigore [21] identified stress of contracting COVID-19 as a reason for being mentally absent from work, rather than physical absence. The constant preoccupation of worrying about getting sick and the consequences of COVID-19 illness during the COVID-19 pandemic creates fear of getting infected with COVID-19. In addition, Shafer et al [27] found that COVID-19 infection decreases the likelihood of working in person compared to other respiratory diseases, indicating the potential difference in perceived risk of COVID-19 infection compared to other illnesses.

Regarding absenteeism, Shafer et al [27] found that individuals with COVID-19 illness were less likely to work compared to individuals with non-COVID-19 acute infections. Grigore [21] discussed more general stress from the pandemic can lead to employees taking absence from work to solve the personal issues they are facing or to focus on improving their mental health. Epidemiological factors influencing absenteeism include social distancing rules at work, stressed more during the pandemic than usual, fear of getting infected at work and transmitting to family, technical support issues, and giving up usual activities to ensure safety. For each category most participants reporting always or often, indicating high stress generated by the pandemic that influences decision on taking an absence from work. 78% of participants felt more stress than usual because of the pandemic compared to before the pandemic. Overall, these pandemic related findings highlighted the significant impact COVID-19 has had on workers physical presence [27], absence [21,27] as well as mental presence [21] in the workplace, implicated in presenteeism and absenteeism behaviour.

## 4 . Discussion

Our review examined definitions of presenteeism and absenteeism in the context of telework and identified antecedents to why they occur. We found that 16 studies provided a definition of presenteeism, while four studies defined absenteeism. In addition, three broad antecedents were identified for both behaviours: organizational, environmental, and individual factors. The interconnected nature of organizational, environmental, and individual factors underscored the complexity of presenteeism and absenteeism in the (tele)workplace. We defined presenteeism and absenteeism as maladaptive and unproductive behaviours that manifested because of policies, environmental, and individual forces that lead to working while sick, or being absent from work, respectively. However, absenteeism is only defined as maladaptive if done excessively or when unwarranted. It is a shared responsibility between organizations and workers to balance these issues during remote work to create positive and productive workplaces.

### 4.1 Definitions for presenteeism and absenteeism

**4.1.1 Overview.** Our review found that most studies defined presenteeism as working while sick, resulting in productivity loss. We found that presenteeism and sickness presenteeism were often used interchangeably, and sickness presenteeism was found to emphasize sickness as the reason for deciding to work online [17,27,38–40,42,43,46]. Additionally, online presenteeism was defined as the behaviour of feeling pressured to be constantly present online for work, which arises during sickness presenteeism, and whilst working remotely respectively [1,21]. For absenteeism, the review revealed consistent definitions across studies, describing it as taking a sick leave from work due to an illness [38,42], being late or leaving early because of an illness [2], and being physically absent from the workplace [21]Absenteeism was broadly understood as "sick leave" or "absence" from work. Based on this review, we defined presenteeism as working while ill, which can result in a loss of productivity, and absenteeism as being absent from work due to sickness.

**4.1.2 Defining presenteeism and absenteeism.** The findings from this review regarding the definition of presenteeism align with previous studies that have defined the phenomenon as working despite being sick, thus reducing overall work efficiency [53]. Our understandings of presenteeism are also consistent with older definitions that defined the phenomenon as "the problem of workers being on the job, but, because of illness or other medical conditions, not fully functioning" [54]. In the broader literature, presenteeism is defined in two ways, influencing how it is measured: as an act of presenteeism (i.e., attending work while ill) or as a consequence (i.e., loss in productivity) of attending work while ill [18]. Additionally, some studies distinguished online presenteeism, which refers to unproductive work behaviours in an online setting [1,21]. Our review found that existing definitions often reflect one of these perspectives, which may explain the variability in how presenteeism is conceptualized. Regarding defining absenteeism, two quantitative studies with the context of skilled trade workers utilized consistent definitions to our study findings: "Lack of physical presence at a given setting and time where there is a social expectation for the employee to be there" [55] and being absent from the workplace [55,56]. A difference to our study findings is that Sichani et al [55] defined absence as no less than 2 hours away from the workplace during scheduled time, which was decided in consultation with construction industry experts [57] emphasized that missing two or more hours of scheduled work time is as damaging to productivity and workflow of the group as missing an entire day of work [55]. Srour et al [56] defined absenteeism as absence from the workplace, which also aligns with our findings of understanding absenteeism as an absence from the workplace, although illness is not specified as the reason for absence. In the skilled trades industry absenteeism is not limited to sick leave, it can be due to any factor deemed by the employers to be excusable, though not specified [55,56]. Srour et al [56] discussed sickness as a plausible reason for absence among construction workers.These study populations did not engage in telework, thus were not a focus of our review [55,56]. Potential gaps within the literature persist regarding absenteeism definitions as there were only four papers that explicitly defined absenteeism, which warrants further investigation.

## 4.2 Antecedents of presenteeism and absenteeism

Our review examined potential antecedents for presenteeism and absenteeism behaviours among teleworkers. Despite varying definitions of presenteeism and absenteeism, similar antecedents were uncovered from the studies. Three themes for antecedents of presenteeism and absenteeism were identified from this review: organizational, environmental, and individual. Organizational factors contributing to presenteeism included elevated job demands and telework [1,17,27,40,42,45,46,48], while lack of access to telework contributed to absenteeism [38]. Regarding environmental factors, distractions such as isolation and interference with familial relations [1,21], as well as poor working environments [48], emerged as contributors to presenteeism behaviours among teleworkers. Meanwhile, environmental factors associated with absenteeism behaviours include types of working environments such as open concept and shared offices which increased probability of contracting sickness from colleagues resulting in absence from the workplace [38]. Individual antecedents for presenteeism included poor mental health [2], feelings of guilt [39,47], and time management issues [1,48]. In contrast, individual antecedents for absenteeism involved job perception [21], COVID-19-related stress [21,27], and general poor mental health [2,42].

Similarities and differences were apparent in the review among the antecedents of presenteeism and absenteeism. Individual factors, such as poor mental health, were found to contribute to both presenteeism and absenteeism behaviours [2,42]. This suggests that interventions targeting poor mental health among teleworkers could help alleviate both presenteeism and absenteeism. On the contrary, organizational factors, such as access to telework, lead to a divergence between two behaviours, highlighting the complexity of telework arrangements. Specifically, the review found that telework access contributed to increased presenteeism [17,27,40,45,46] while workers without telework access were more likely to exhibit absenteeism behaviours [38]. Thus, teleworking can help reduce absenteeism, it may simultaneously promote presenteeism. This interconnectedness highlights the importance of addressing both behaviours holistically with organizational strategies. This can include the need for organizations to improve workload distribution, ensuring realistic expectations and to promote work-life balance through flexible work arrangements. Additionally, creating a supportive and inclusive work environment as well as providing resources, such as time management tools, can also guide employees to manage their workload and work environment effectively, reducing the risk of both behaviours. Moreover, organizations should establish sick leave policies to guide teleworkers on when it is appropriate to take sick leaves. This may help mitigate the guilt some employees experience when taking a sick leave while working remotely.

These findings on telework also have important implications for equity, especially in terms of gender and socioeconomic status. Presenteeism may be more common among women who have caregiving responsibilities and feel pressured to work despite being ill, as well as lower-income employees who lack the necessary resources for a home office, such as a quiet workspace or proper technology. Organizations should address these differences, ensuring equitable access to resources and support for all employees, regardless of gender or socioeconomic status.

Previous studies on the antecedents of presenteeism and absenteeism yielded similar results to our review. For instance, factors such as high job demand, poor mental health, and job dissatisfaction were found to contribute to both behaviours among both teleworkers and non-teleworkers [57,58]. Interventions targeting these factors could help improve outcomes for all workers, regardless of their working arrangements. Moreover, factors such as access to teleworking, working environments, and COVID-19-related stressors are unique challenges faced by teleworkers. For that reason, organizations that provide teleworking options for their workers should consider these factors and invest in interventions that tackle them. Doing so can not only improve teleworkers' health, but also bolster teleworkers' productivity.

## 4.3 Social determinants of workers health: Relationship between organizational, environmental, and individual antecedents

Organizational, environmental, and individual antecedents impact presenteeism and absenteeism behaviour. Organizational factors, such as policies, could impact the work environment, which can ultimately impact employees' decision to engage in presenteeism or absenteeism behaviour. This understanding can be drawn from Dahlgren and Whitehead's

[59] social determinants of health framework, where individual-level health and wellbeing is impacted by environmental and organizational factors at the meso- and macro-level, respectively [60]. Through applying the SDH framework, we can improve our understanding of how population-level determinants – such as organizational policies on work hours, flexibility, and occupational health – create working conditions that either mitigate or exacerbate presenteeism and absenteeism behaviours. Prior research has demonstrated that organizational policies influence work environments [59], while environmental factors, such as physical workplace conditions, affect individual outcomes like job productivity [61]. Furthermore, organizational work policies impacted individual health outcomes including mental health [60], infection risks [60], work experiences [62], and employee productivity [62]. It is important to note that the cascading impact of organizational factors on environmental and individual level antecedents to presenteeism and absenteeism are not unidirectional but rather bidirectional, wherein health and job-related factors can impact and are impacted by social and environmental conditions, as shown in the social determinants of health model [59]. The lines between environment and organizational antecedents are often blurred; with some studies merging both while others distinguishing one from the other, and so does the social determinants of health model where institution-level and environmental determinants coincide. For example, some organizations used the Job Demands and Job Resources model to shelter their teleworkers from the stressors of the pandemic, while the workers also used their individual means for self-recovery [63]. For clarity in our study, we categorized organizational factors (e.g., such as policies) to be separate from environmental factors (e.g., work environment) as policies can create the environment in which one works and ultimately impact individual health and wellbeing based on the social determinants of health model [59] where the individual impact in our study is teleworkers and their presenteeism and absenteeism behaviour. In this context, presenteeism and absenteeism are therefore conceptualized as the downstream impacts of organizational and environmental conditions that create unfavourable working environments, resulting in maladaptive and unproductive behaviour in the workplace. The SDH framework thus provides a comprehensive lens through which to analyze these behaviours, underscoring the need for workplace policies that foster healthy, supportive, and sustainable work environments.

## 4.4 Limitations, recommendations, and future directions

**4.4.1 Limitations.** There are a few limitations to this review. All the studies were conducted in Europe and North America, regions which are dominated by High-Income Countries. This geographical concentration suggests a potential bias towards perspectives and experiences from the Global North, with a dearth of research from the Global South where Low- and Middle-Income Countries are more prevalent. As such, there are limitations to the generalizability of this study as it does not capture the full range of perspectives from other under-researched countries nor explore content written in other languages. In the Global South, there are various factors that can shape telework experiences that may not be reflected in the existing literature. For example, differences in access to technology and societal norms may impact how presenteeism and absenteeism occur in these regions. To address this imbalance, future reviews should include non-English articles to provide a more comprehensive understanding of presenteeism and absenteeism across diverse contexts. Additionally, due to the lack of research from the Global South, future research should conduct more studies in these regions to address this knowledge gap. Further, some articles identified in this review did not contain enough information on definitions related to presenteeism and/or absenteeism nor antecedents of it, which made interpretation of the results difficult to assess. Lastly, there were only four out of 18 articles that included a definition of absenteeism. Due to the low number of studies that reported this phenomenon, this may limit the ability to conceptualize a standard definition of absenteeism. Moving forward, a separate literature search can be conducted for absenteeism.

**4.4.2 Recommendations.** Our study found that presenteeism and absenteeism during telework could result from a lack of social and coping resources and support that would have otherwise been obtained from in-person work interactions [1]. To address this, employers should ensure that teleworkers are effectively trained during onboarding in the first few weeks of orientation on how to navigate the virtual workspace, including how to access social supports,

and provide general resilience strategies to support virtual work. In addition, Adisa et al [1] found a factor related to job perception and working online from home is difficulty in balancing work and life, and a general poor adaptation to work from home environments [1]. Employers could consider preparing employees for telework arrangements during onboarding. Resources such as training and work from home tips can be provided to new employees to ensure an effective transition from in person experience to virtual experience. This could also help address job dissatisfaction as an antecedent to presenteeism and can bolster employee productivity. Moreover, stress management and workplace mental health interventions from a top-down organization level to employee individual-level like cognitive behavioural therapy insurance coverage, meditation, and mindfulness breaks during working hours show promise for addressing poor mental health and productivity in the workplace [64]. Additionally, providing opportunities for both virtual and in person social connections could address loneliness that comes with working in a virtual environment.

**4.4.3 Future directions.** This scoping review has identified several avenues for future research to address the limitations of the current review. Notably, the majority of studies included in this review were conducted in Western, high-income countries, highlighting a gap in the understanding of the presenteeism and absenteeism behaviours of teleworkers in the Global South. Future research should explore these behaviours within diverse cultural and geographic contexts to develop a more comprehensive framework for understanding the determinants of presenteeism and abstenteeism across different settings. Moreover, research on absenteeism among teleworkers remains limited, with few studies examining its underlying factors. Given this gap, future studies should priortize investigating the relationship between telework and absenteeism to generate more rigorous findings and inform strategies for mitigating absenteeism in telework settings. Additionally, since telework may be suitable for some workers to achieve work-life balance, it would be desirable to see future research on how employers can appropriately manage, and structure telework to be conducive to a work-life balance for employees. Lastly, teleworking can help increase employment opportunities for individuals with disabilities, making it essential for future research to examine best practices in this area [65].

# 5. Conclusion

Overall, this scoping study conceptualized definitions of absenteeism and presenteeism in the context of telework and identified three interconnected antecedents – organizational, environmental, individual – through the lens of the social determinates of health network. Presenteeism and absenteeism can be thought of as maladaptive behaviours in the workplace if left unaddressed. Maintaining mental health and safety is crucial, as it affects workplace interactions and management practices. It is important for organizations to adopt positive approaches to mental health and safety to allow companies to better recruit and retain employees, improve engagement and productivity, while also reduce workplace issues such as conflict, turnover, and absenteeism. Stress management programs are essential to promote adaptive coping mechanisms and can help tackle the antecedents of poor mental health linked to these behaviours.

These findings have profound implications, as policy makers and organizations could tailor telework policies to address absenteeism and presenteeism in organizational contexts, such as small businesses and multinational corporations. For small businesses, which often operate with limited resources and fewer employees [66], telework policies should focus on fostering flexibility while ensuring productivity. Given that small businesses may lack formal HR structures, policies could potentially target workload management, technology access and support, and mental health support. In contrast, multinational corporations have more widespread and diverse workforce needs [67]. As a result, telework policies should consider comprehensive well-being programs, equitable technology access and support, and structured flexibility.

Furthermore, conceptualized definitions of presenteeism and absenteeism could develop consistent measures that help identify the phenomenon throughout research. Our study defines presenteeism as working while sick, and absenteeism as being absent from work. Both phenomena can be influenced by organizational, environmental, and individual factors. These findings can improve workplace efficiency and address health and wellbeing issues in remote work. Identifying

antecedents enables the development of safeguards to prevent these behaviours, fostering more efficient and supportive telework environments.

## Supporting information

**S1 Data. Seach history for each database.**
(DOCX)

**S2 Data. Tables of included and excluded studies with rationale.**
(DOCX)

## Author contributions

**Conceptualization:** Behdin Nowrouzi-Kia, Sharada Nandan, Edris Formuli, Kishana Balakrishnar, Ali Bani-Fatemi, Aaron Howe, Yiyan Li, Luke A. Fiorini, Shane Avila, Chantal Atikian, Kathy Zhou, Mahika Jain, Basem Gohar.

**Data curation:** Sharada Nandan, Edris Formuli, Kishana Balakrishnar, Yiyan Li, Shane Avila, Chantal Atikian, Kathy Zhou, Mahika Jain.

**Supervision:** Behdin Nowrouzi-Kia.

**Writing – original draft:** Sharada Nandan, Edris Formuli, Kishana Balakrishnar.

**Writing – review & editing:** Behdin Nowrouzi-Kia, Sharada Nandan, Edris Formuli, Kishana Balakrishnar, Ali Bani-Fatemi, Aaron Howe, Yiyan Li, Luke A. Fiorini, Basem Gohar.

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
