## [Decision Letter · Decision Letter 0]

21 Aug 2024

PMEN-D-24-00187

Sick leave or Work Sick? Examining the Antecedents and Conceptualizations of Presenteeism and Absenteeism among Teleworkers During COVID-19: A Scoping Review

PLOS Mental Health

Dear Dr. Nowrouzi-Kia,

Thank you for submitting your manuscript to PLOS Mental Health. After careful consideration, we feel that it has merit but does not fully meet PLOS Mental Health’s publication criteria as it currently stands. Therefore, we invite you to submit a revised version of the manuscript that addresses the points raised during the review process.

We look forward to receiving your revised manuscript.

Kind regards,

Bochra Nourhene Saguem, M.D.

Academic Editor

PLOS Mental Health

Journal Requirements:

Additional Editor Comments (if provided):

Reviewers' comments:

Reviewer's Responses to Questions

**Comments to the Author**

1. Does this manuscript meet PLOS Mental Health’s publication criteria ? Is the manuscript technically sound, and do the data support the conclusions? The manuscript must describe methodologically and ethically rigorous research with conclusions that are appropriately drawn based on the data presented.

Reviewer #1: Yes

Reviewer #2: Partly

2. Has the statistical analysis been performed appropriately and rigorously?

Reviewer #1: I don't know

Reviewer #2: No

3. Have the authors made all data underlying the findings in their manuscript fully available (please refer to the Data Availability Statement at the start of the manuscript PDF file)?

Reviewer #1: No

Reviewer #2: Yes

4. Is the manuscript presented in an intelligible fashion and written in standard English?

Reviewer #1: Yes

Reviewer #2: No

5. Review Comments to the Author

Reviewer #1: 1 The purpose of this paper's research is to define presenteeism and absenteeism in the (tele)workplace and to systematically identify the factors contributing to their occurrence.

2 The method is appropriate as it extracts definitions of presenteeism and absenteeism in the (tele)workplace during COVID-19 from seven online databases.

3 On the other hand, the researchers consulted experts to classify the antecedent factors of presenteeism and absenteeism, but it is difficult to understand how they were classified, so I would like to see them summarized in a table.

4 I don't think it's new to conclude that antecedent factors at the organizational, environmental, and individual levels interact with each other. In the next study, it is hoped that statistical methods will be used to further analyze the content of individual factors.

5 It is stated that there were few papers that reported the definition of absenteeism, but this is not surprising since COVID-19 has led to an increase in people working from home and presenteeism, where people work online instead of being absent from work. It would be necessary to mention that.

6 Reading the entire paper, absenteeism was a problem before the COVID-19 outbreak, but during the COVID-19 outbreak, telework is required and presenteeism has increased, and presenteeism is undesirable from the standpoint of worker health and work efficiency. I thought this would be a consideration.

7 This paper shows that telework due to the COVID-19 pandemic has resulted in various problems as people have been forced to work from home without a proper work environment. Since telework may be suitable for some workers to achieve work-life balance, it would be desirable to see future research on how employers can appropriately manage telework.

Reviewer #2: Although I have recommended major revision, I do NOT want to discourage the authors from revising and re-submitting their manuscript. The study explores significant shifts in the world of work since the onset of Covid-19 and offers insights on presenteeism and absenteeism which are useful to employers and practitioners alike. However, the manuscript needs a good deal more work before being publication-ready as it is overly wordy, somewhat disorganised in its structure and contains a high level of errors (many minor but some more major). My overarching impression is that the authors “ran out of steam” as the writing process went on, resulting in inadequate proofreading and cross-checking and a lack of tight editing. Dispersed responsibilities due to a relatively large author group can contribute to this.

In no particular order, my recommendations are as follows.

1. Given the specialist focus of the journal, the authors should more overtly link their core focus (antecedents of presenteeism and absenteeism amongst teleworkers) to mental health. This could be achieved through insertion of a sentence or two in the opening section, with relevant citations, as well as in the discussion and conclusion sections.

2. Since the Social Determinants of Health model is ultimately used to frame the findings of the review, this model should also be introduced briefly in the opening section of the manuscript or alternatively referred to in the methods section.

3. Consistent and logical word order for key phrases would be easier on the reader’s eye and brain. For instance, the manuscript switches between “absenteeism and presenteeism” and “presenteeism and absenteeism”. A conventional alphabetical strategy could be used (a before p), alternatively the phenomenon which “showed up’ most strongly in the findings could be put first (from lines 597-598 it seems that presenteeism was generally more visible – at least in those studies which provided definitions). Also be consistent in how past, present and future tense is used – for instance, describe your methodology – what you did – using past tense and when reporting the findings of the reviewed studies and referring to the broader literature, settle on either past tense (“as Smith 2007 noted...”) or present tense (“as Smith 2007 note...). Also be consistent in how you refer to the studies – Table 2 uses “... et al” but in the body of the manuscript you sometimes refer to “... et al” and sometimes to “... and colleagues”.

4. Terms should be carefully chosen – for instance, avoid using “concept” (lines 153-154) given that this word is already present in the abbreviation PCC. Rather use something like “construct”, “component” or “principle”. Also, be consistent – for instance, you mostly refer to the items you reviewed as “studies”, but on your PRISMA-ScR diagram you refer to “reports” and in line 240 you use the term “articles”. In lines 302-304 you use “three major topics” – you could simply say “we identified three types of antecedents, namely organisational, environmental and individual”. Both in the abstract and later in the manuscript, also make it clear how you define home environment and personal environment, if you view these as different from (or overlapping with) the work environment.

5. Reduce wordiness by removing superfluous words. For instance, in line 162 replace “noted how this will potentially...” (5 words) with “noted would potentially...” (3 words). There are also places where you could collapse two sentences into one – eg: lines 434-435 and 435-438. Shorten sentences where possible and avoid meaningless terms like “and so on” – see for instance lines 304-305 where you could trim the first sentence down to “Organizational antecedents included factors such as policies, job demands and workplace compensation”. Similarly, your sentence on lines 550-551, “Previous studies have investigated the antecedents of presenteeism and absenteeism for workers which have yielded similar results to this review” (20 words) could be shortened to “Previous studies investigating antecedents of absenteeism and presentations yielded similar results to our review” (14 words).

6. Timeframe your review precisely and unambiguously. Referring to “between 2000-2023” in line 161 and “studies between the years 2019 to 2023” in line 164 is contradictory and confusing. Rather say “studies published from 2000 to 2023 (inclusive)” or, even better, “studies published from January 2019 to December 2023 (inclusive)”. Also make absolutely clear that the inclusion criterion was the publication date of the study, not the data collection date.

7. The Methodology section is rather confusing, clumsy and circular, making it hard for the reader to picture how (and why) you went about your study. For instance, there is no explanation of why you included gender breakdowns but not age breakdowns. There is also little information on what analytical methods were used – it would be helpful to add a sentence like “Simple numerical counts and percentages were utilised to explain the studies’ geographic settings, methodologies, sample sizes and (where available) gender breakdowns. Frequency counts revealed the prevalence of presenteeism and absenteeism definitions, respectively”.

8. It is a little difficult to understand the roles of the research team members, particularly in relation to BNK – described in line 177 as “an occupational therapy expert” but in line 196 as “an expert in occupational health” (occupational therapy and occupational health being two different disciplines). Also, talking about “consulting an expert” (pg 36) creates a sense of having obtained external input when instead the consultation process seems to have been internal, drawing on knowledge within the research team. External consultations may be used to add trustworthiness and credibility, but would not merit author status – instead being reflected in the Acknowledgements section.

9. Keep your language as crisp and clear as possible when outlining your inclusion and exclusion criteria. Your last inclusion criterion in Table 1 (“Studies that implemented telework...”) should probably instead be “Studies that examined telework....”, as presumably the researchers did not implement telework themselves but instead researched how it was implemented by others. Likewise, your last exclusion criterion in Table 1 should probably read “Studies which examined in-person work environments and/or telework implemented before 2019”, to match the body of the text (lines 148-149).

10. In general, there is some muddling of the Results section (which should simply lay out the findings) and Discussion section (which should pull the findings together and make meaning out of them). Giveaways that one has veered into discussion within the results section are words like “interestingly”, “strikingly”, “It is plausible to suggest that...” or “...should be further explored in future studies”. Line 223 in the results section (“gaps were identified in definitions”) can also be deferred to the discussion. In the discussion section itself, it would be helpful to quantify the findings – eg: by mentioning the actual number (and percentage) of studies which reported a particular finding, which included a particular definition, etcetera. Also be very clear with regards to when you are mentioning the reviewed studies and when you are referring to the broader (non-reviewed) literature. In lines 344-347 it makes no sense to use the term “previous studies” in referring to other studies which were published within the same timeframe as the reviewed studies – find another way to describe them.

11. I find your PRISMA diagram at odds with the text – please re-check it carefully. At the first step, your diagram shows 1,117 records retrieved but in the text you refer to 1,116, so from then onwards the diagram and text do not match. You say you ultimately reviewed 21 studies but doing the maths through the PRISMA diagram suggests that there were 25 studies identified for inclusion (146-121=25); even given that three were belatedly removed, that would still leave 22, not 21? Yet your Table 2 clearly shows that only 21 studies were analysed. Please re-check all your figures for each step of the diagram. With regards to the three studies which were removed only after data extraction took place (lines 229-232), why was this so noteworthy? This kind of thing does happen during systematic reviews and once team consensus is reached that a study is not eligible after all it simply moves to the box showing all excluded studies and the grounds for exclusion.

12. Your including a study based on secondary data (Navarro et al) is confusing given your inclusion criterion of empirical studies only. Should that study not have been excluded on grounds of “wrong study design”? If it contained useful information such as definitions of presenteeism and/or absenteeism, this could feature in the discussion section where you relate your findings to the broader literature.

13. With regards to the Results section, sub-heading 3.1 (Descriptive Findings) could be replaced with “Search Results”. Section 3.2 (Study Characteristics) could give a much clearer descriptive breakdown of the studies than it currently does. Make sure that this ties in clearly with Table 2, after first resolving exactly how many studies were analysed (ie. whether Navarro et al should have been included) and then determining the best way to organise the table. Logical approaches are to list the studies in chronological order, to list them alphabetically, or to cluster them by methodologies and, within each cluster, using sub-ordering by chronology or alphabet. The accompanying text should provide additional information for readers to understand the results, with percentages added to absolute numbers to bring the results more alive and again using logical sub-ordering. For instance [using what is currently shown in your Table 2]:

• You could describe the geographic characteristics of the studies like this:

“Only 2 (9.5%) of the studies had a multi-country focus (one being focused on 28 European countries and the other having a global focus), with the other 19 (90.5%) having a single-country focus. Amongst the latter, the most represented countries were the United Kingdom (4 studies) followed by Germany, Japan and the United States (2 studies each). The remaining countries, each represented by a single study, were Austria, Canada, Lithuania, Norway, Portugal, Romania, South Korea and Spain. In all, the North American and European regions were dominant, accounting for 20 (95.2%) of the studies.”

• You could describe the methodological characteristics of the studies like this:

“From a methodological point of view, most of the studies (16; 76.2%) were quantitative, with 9 (56.3%) of those being cross-sectional, 3 (18.8%) being cohort studies, one (6.3%) each being, respectively, a secondary data analysis, a daily diary study, an experimental study and a study which did not specify what precise method was used. Three (14.3%) of the studies were qualitative, with 2 (66.7%) of these involving semi-structured interviews and one (33.3%) not specifying the precise method used. The remaining two studies (9.5%) were mixed-method studies.

• You attempted to describe the gender breakdown of the participants across all the

studies but this is not very clear – for instance, why is it “important to note that Raisiene and colleagues (2023), a Lithuanian study, had a sample comprising only 26.2% female”? And why do you include studies with 50% females amongst those “with a female minority”? You could perhaps say something like this:

“Most of the studies (18; 85.7%) provided gender breakdowns, with female representation ranging from 26.2% to 95.8% with a median value of ....%. However, of the total participant set of 54,457 across all the study samples, no gender was specified for 14,058 (25.8%)”.

14. Be careful about ascribing cause where none may exist – thus, instead of saying “to identify antecedents for why...” (line 301) (my emphasis) it would be safer to say “antecedents to the occurrence of...”. As I understand it, an antecedent is simply something that existed or happened before another thing and is not always the cause or origin of the other thing, even though the two things have some logical relationship. In lines 348 and 357 you use the term “contributed to...” – did the reviewed studies use that term? Or did they just find an apparent association between two phenomena, without assigned causality? In lines 387-392 it is unclear whether the review authors are reporting their own views or reporting on the findings of the reviewed studies.

15. In the Discussion section I suggest replacing the terms “Western bias” and “non-western countries” when discussing the geographical limitations of the review. Rather use more contemporary terms like Global North and Global South, or refer to most of the studies having been carried out in High-Income Countries rather than Low and Middle Income Countries. It is important to make clear what your own recommendations are, emanating from your study, rather than just referring to recommendations made by other authors. The Conclusion section should be far shorter, summing up the essence of the review in a few sentences rather than repeating aspects of the discussion and recommendations. The very last sentence, in using the phrase “...for teleworkers and non-teleworkers alike” (line 670), actually seems to dilute the whole purpose of the review and leave the reader with a sense of incompletion.

16. Finally, with regards to proofreading and error correction, here are some pointers:

a. Whilst re-checking and finalising Table 2 pay attention to fine points like formatting – for instance, under the Sample Size column you have 275 next to Biron et al, rather than n=275. You also use the format n=x in some places and n = x in others.

b. Watch out for these typographical and grammatical errors:

• Word omissions – eg: the word “to” seems to have been left out on line 79 (“...due [to] working while sick”).

• Incorrect grammar – eg: “physical present” on lines 80 and 81 should read “physically present”; “being physically absent to work” in line 82 should probably read “being physically absent from work”; “our eligibility criteria was modified” in line 160 should read “our eligibility criteria were modified”; “Data charting and data reporting was performed” in line 181 should read “Data charting and data reporting were performed”.

c. Missing or inconsistent punctuation and numbering – lack of full stops in various places; different ways of using the abbreviation of ‘for example” (shown as e.g. in some places and e.g., in others); a missing apostrophe in line 104 (“...workers’ ill-health”); missing question marks for the research questions on lines 105-107; missing comma in line 127; the sub-heading “Data Synthesis” should be numbered “2.5 Data Synthesis”.

• Misspelling of significant words - eg: ‘PRIMA-ScR’ instead of ‘PRISMA-ScR’ in line 120; ‘preformed’ instead of ‘performed’ in line 370; ‘presentism’ instead of ‘presenteeism’ in line 640.

• Strange symbols appearing in ref 9 on the reference list.

• Clumsy wording – eg: a sentence in lines 150-151 would be shorter and read better if worded “...we wanted to ensure that organizations that implemented telework did so due to COVID-19”.

d. Avoid repetition and wordiness, for example:

• The word “overall” is used to start two close sentences on page 2 (lines 15 and 19).

• In lines 47 and 48 there appears “...after the height of the COVID-19 pandemic restrictions-COVID-19 restrictions...”.

e. Referencing errors and inconsistencies need to be identified and corrected, for example:

• ‘Tremblay’ is shown as a source in line 54 but does not appear in the reference list.

• Inconsistent referencing in relation to Ruhle makes it difficult to know whether the author cited in line 60 was actually Ruhle & Schmoll (no. 18 on the reference list) or Ruhle as shown in reference 16 on the reference list).

• Single author Grigore (ref 23 on the reference list) is referred to as “Grigore and colleagues” in various places in the body of the manuscript. Reference 23 also seems incomplete, ending with “8:”. On line 293 you need to add [23] after mentioning Grigore and before citing Cooper and Locke.

• In lines 417-418 you refer to Brosi and colleagues, but ref 41 shows only two authors, Brosi & Gerpott.

• Different dates shown for Shafer and colleagues – shown as 2023 in lines 453-454 but Table 2 (pg 16) and ref 17 gives the date as 2020.

• There is a mismatch in dates for Arksey & O’Malley – given as 2007 in the body of the manuscript (lines 113, 170 and 182) but as 2005 in ref 28.

• There is incorrect spelling of Laranjeira/Laranjira, between the text and the reference list.

• The verbatim quote from Cooper & Locke in lines 293-295 should get a page number if possible.

• In line 171 you mention Zotero but you provide no reference for this, in the way you have done for Covidence.

6. PLOS authors have the option to publish the peer review history of their article (what does this mean? ). If published, this will include your full peer review and any attached files.

**Do you want your identity to be public for this peer review?** For information about this choice, including consent withdrawal, please see our Privacy Policy .

Reviewer #1: No

Reviewer #2: No

---

## [Decision Letter · Decision Letter 1]

29 Sep 2024

PMEN-D-24-00187R1

Sick leave or Work Sick? Examining the Antecedents and Conceptualizations of Presenteeism and Absenteeism among Teleworkers During COVID-19: A Scoping Review

PLOS Mental Health

Dear Dr. Nowrouzi-Kia,

Thank you for submitting your manuscript to PLOS Mental Health. After careful consideration, we feel that it has merit but does not fully meet PLOS Mental Health’s publication criteria as it currently stands. Therefore, we invite you to submit a revised version of the manuscript that addresses the points raised during the review process.

We look forward to receiving your revised manuscript.

Kind regards,

Bochra Nourhene Saguem, M.D.

Academic Editor

PLOS Mental Health

Journal Requirements:

Reviewers' comments:

Reviewer's Responses to Questions

**Comments to the Author**

1. If the authors have adequately addressed your comments raised in a previous round of review and you feel that this manuscript is now acceptable for publication, you may indicate that here to bypass the “Comments to the Author” section, enter your conflict of interest statement in the “Confidential to Editor” section, and submit your "Accept" recommendation.

Reviewer #1: All comments have been addressed

Reviewer #2: All comments have been addressed

2. Does this manuscript meet PLOS Mental Health’s publication criteria ? Is the manuscript technically sound, and do the data support the conclusions? The manuscript must describe methodologically and ethically rigorous research with conclusions that are appropriately drawn based on the data presented.

Reviewer #1: Yes

Reviewer #2: Partly

3. Has the statistical analysis been performed appropriately and rigorously?

Reviewer #1: Yes

Reviewer #2: I don't know

4. Have the authors made all data underlying the findings in their manuscript fully available (please refer to the Data Availability Statement at the start of the manuscript PDF file)?

Reviewer #1: Yes

Reviewer #2: Yes

5. Is the manuscript presented in an intelligible fashion and written in standard English?

Reviewer #1: Yes

Reviewer #2: No

6. Review Comments to the Author

Reviewer #1: I have tried to understand the content of the paper and comment on it. As you say, I believe that the quality of the paper improved by making revisions in response to the reviewers' comments.

Reviewer #2: REVIEWER COMMENTS

Firstly, I commend the authors for not being deterred by the request for major revisions and acknowledge their willingness to make multiple changes to their manuscript in line with my recommendations – and the speed with which they did so.

Although I did not previously comment on the use of keyword ‘organisational psychology’ in the first manuscript, I note and agree with this having been changed to ‘organisational health’.

Introductory section

I thank the authors for strengthening the mental health framing and making reference to the social determinants of health model, as requested.

Methodology section

I thank the authors for refining this section considerably, including in relation to their use of terminology.

As per my previous comment, “studies that moved to telework” (line 154) should read “studies that examined a move to telework...” or “studies that focused on a move to telework”. The studies themselves did not implement telework.

There are still some places where the ordering of information is a bit confusing – for example, lines 201-203 seem to pre-empt the explanation of how categories were developed, which appears a couple of paragraphs later; the authors could move those lines to there. They could also shorten “...would be categorized under the category ‘individual antecedents’” to the more reader-friendly “...would be categorized under ‘Individual Antecedents’’’. In line 209, it may be helpful to refer to ‘the data corpus’ (meaning the body of literature analysed during this particular review) or just ‘the studies’ – using the term ‘the literature’ here is confusing, as it brings to mind the broader framing literature referred to in the Introduction and Discussion sections. See also line 414 – rather use “the studies”.

With regards to the authors’ “consulting” of experts, for maximum transparency I recommend an even clearer explanation that they were within the research team (lines 211-212); for instance, “The research team drew on internal expertise for this, in particular that of B.N.K (an expert in occupational health) and that of Y.L (an experienced social health researcher”.

Results section

I acknowledge the extensive work the authors have done here, reporting their findings more clearly and removing text which belonged more appropriately in the Discussion section. With regards to the PRISMA-ScR diagram and the text accompanying it, I thank them for clarifying that they excluded a grey literature item from the outset at the first stage. They should make this crystal clear by changing the extra sentence added in Lines 237-238 to explain this, and expanding the top right box on the PRISMA diagram to say ‘Duplicate articles removed (n=289)’ AND ‘Grey literature item removed (n=1). Then the rest of the numbers will make sense.

The authors’ response to my comment about the timeframe of the review raises a concern regarding the inclusion of the 2019 study of Hu et al. I exercised lenience previously due to the possibility of it being a late 2019 study, as the reference list did not specify the month of publication. However, I’ve now sourced the study and seen that it was published in March 2019, having first been submitted in August 2018 – thus, although it does not specify the date on which the survey it analysed was carried out, that could not have been more recent than 2018. I know the review utilised publication date not data collection date, but I’m concerned that this study seems to go against the authors’ Exclusion criterion in lines 156-157 – thus, its inclusion may well be queried by critical readers. I’m not suggesting they re-do the analysis, but I would like to see reflected how the research team approached this particular study – for instance, if they debated whether or not to include it, it would be useful to know the reasoning behind the fact that they did include it for analysis, rather than just referring to it within the broader literature relevant to their topic.

When it comes to reporting results, I suggest to the authors that many readers prefer percentages to be included as those help make the findings come alive and require few extra characters. They do not need to keep mentioning the 21 studies each time – thus, just as one example, “Out of 21 studies, a total of 17 studies...” (line 280) could be shortened to “Seventeen (81.0%) of the studies....”. I urge the authors to follow the convention of writing out in word form all numbers below 10 (eg. four instead of 4). They should also decide how many decimal places to use and be consistent in that throughout the manuscript. In my view, 81.2% and 80.9% mean pretty much the same thing, and rounding off each to 81% is easier on the reader’s eye.

Discussion section

The Discussion section is much improved and flows better now. I remain with a few concerns, though. For instance, the use of “In research” (line 536) is confusing – I presume the authors are referring to “In the broader literature...”, but it would be better to say so clearly. Other than this, it is generally much clearer when they are referring to their own data corpus and when they are referring to the broader literature.

I thank the authors for switching to the terminology “Global North “and “Global South”, an arguably more contemporary way of referring to developed versus developing countries. While they correctly acknowledge the review’s “potential bias towards perspectives and experiences from the Global North” (which they have noted includes many High-Income Countries) they may consider replacing “....there is a significant gap in research from the Global South and other under researched regions” with the more explicit “...from the Global South and in particular from Lower- and Middle-Income Countries”.

The sentence the authors have added in lines 643-645 is rather confusing – ie. their comment that “The low search yield on papers exploring presenteeism is not surprising since COVID-19 has led to an increase in people working from home and engaging in presenteeism...”. Surely an increasingly prominent phenomenon tends to trigger increased research on it? Or perhaps the authors meant to say that it is as yet too recent a phenomenon to have been subjected to much research? Rather confusingly, in lines 679-680 the authors also comment on scarcity of research on absenteeism amongst teleworkers – which suggests that there’s a general lack of both presenteeism and absenteeism research.

The authors’ added sentence in lines 682-683 (“Lastly, since telework may be suitable for some workers to achieve work-life balance”) could perhaps be expanded by adding in a mention of how telework can also help increase access to work for people with disabilities. Ideally, they should cite a study that reflects this, if they came across one.

Conclusion section

The reworked Conclusion is better but still notably long, repeating a bit much detail from earlier parts of the manuscript. I also note that, conventionally, a conclusion does not cite any authors as its purpose is to draw to a conclusion the authors’ original contribution.

I advise using the broader term “mental health” (in line 697 and elsewhere) rather than “psychological health”, which is arguably a sub-category of mental health and has not been a big focus of the paper.

I suggest ending with punchy, shorter sentences rather than the long one the authors have used. This could be shortened and broken up something like this:

Identifying antecedents of presenteeism and absenteeism allows safeguards to be

developed to prevent these phenomena from occurring during telework. In turn, this can foster more efficient and

supportive workplaces.

Reference list

I have not had time to check each and every citation and reference list entry, but I do note that the authors seem to have made relevant numbering corrections after shifting around text and citations. They should do a final cross-check.

Other comments

My trained proofreader’s eye could not help but notice persistent typos and inconsistencies, which I think are worth correcting to really take the manuscript to the next level. This is especially important given that the journal does not provide internal proofreading and editing.

• Re-check for consistent use of the abbreviation of ‘for example” (shown as e.g. in some places and e.g., in others) – see line 57 for instance, and in the Exclusions column of Table 1.

• Proofread for missing punctuation – eg: see missing full stop in line 59, missing comma before “absenteeism” in line 131, missing apostrophe after “teleworkers” in line 594

• Fix incorrect spacing – eg:

o lack of a space after [19] on line 63

o lack of an empty line between sections 2.2 and 2.3 and sections 2.4 and 2.5

o lack of an empty line before section 3

• Identify and correct grammatical, tense and other errors – like:

o line 75: “are prominent reason for” should read “are a prominent reason for”.

o line 83: “a reason for being mentally absent but physically present from work” should read “a reason for being mentally absent from but physically present at work”

o line 143 – “was” should be “were”

o lines 148 and 149 – “discuss” should be “discussed”

o lines 206-207 repeat an earlier sentence.

o line 228 – “was” should read “were”

o line 231 – “computed” would make more sense than “conducted”

o line 284 – should read “Table 3”, not “tables 3”

o lines 311 and 319 – an “s” has been added to “understand” but the first version was correct as the studies mentioned both had more than one author

o line 392 – “was associated with” should read “were associated with”

o line 480 – use the past tense: “Grigore identified stress....”

o line 585 – “absenteeism and presentations” should presumably read “absenteeism and presenteeism”

o line 605 – please correct “couldan”

o lines 684-685 – “conducive of a work-life balance” should probably read “conducive to a work-life balance”

7. PLOS authors have the option to publish the peer review history of their article (what does this mean? ). If published, this will include your full peer review and any attached files.

**Do you want your identity to be public for this peer review?** For information about this choice, including consent withdrawal, please see our Privacy Policy .

Reviewer #1: No

Reviewer #2: No

---

## [Decision Letter · Decision Letter 2]

19 Nov 2024

PMEN-D-24-00187R2

Sick leave or Work Sick? Examining the Antecedents and Conceptualizations of Presenteeism and Absenteeism among Teleworkers During COVID-19: A Scoping Review

PLOS Mental Health

Dear Dr. Nowrouzi-Kia,

Thank you for submitting your manuscript to PLOS Mental Health. After careful consideration, we feel that it has merit but does not fully meet PLOS Mental Health’s publication criteria as it currently stands. Therefore, we invite you to submit a revised version of the manuscript that addresses the points raised during the review process.

We look forward to receiving your revised manuscript.

Kind regards,

Bochra Nourhene Saguem, M.D.

Academic Editor

PLOS Mental Health

Journal Requirements:

Additional Editor Comments (if provided):

Reviewers' comments:

Reviewer's Responses to Questions

**Comments to the Author**

1. If the authors have adequately addressed your comments raised in a previous round of review and you feel that this manuscript is now acceptable for publication, you may indicate that here to bypass the “Comments to the Author” section, enter your conflict of interest statement in the “Confidential to Editor” section, and submit your "Accept" recommendation.

Reviewer #1: All comments have been addressed

Reviewer #2: All comments have been addressed

2. Does this manuscript meet PLOS Mental Health’s publication criteria ? Is the manuscript technically sound, and do the data support the conclusions? The manuscript must describe methodologically and ethically rigorous research with conclusions that are appropriately drawn based on the data presented.

Reviewer #1: Yes

Reviewer #2: Partly

3. Has the statistical analysis been performed appropriately and rigorously?

Reviewer #1: Yes

Reviewer #2: No

4. Have the authors made all data underlying the findings in their manuscript fully available (please refer to the Data Availability Statement at the start of the manuscript PDF file)?

Reviewer #1: Yes

Reviewer #2: Yes

5. Is the manuscript presented in an intelligible fashion and written in standard English?

Reviewer #1: Yes

Reviewer #2: No

6. Review Comments to the Author

Reviewer #1: (No Response)

Reviewer #2: As with my first review, I hope NOT to deter the authors from making further revisions in order to make their manuscript (in my opinion) publishable, as the subject matter remains valuable and helps fill a knowledge gap. I also acknowledge the efforts put in to respond to the first review and the re-review, but the impression gained is that the second revision of the manuscript was rushed and not sufficiently thorough. Thus, many of the same concerns remain regarding typographical errors, tense inconsistencies, formatting and grammatical errors. There is now also a new set of concerns introduced by the revision of the data corpus from 21 to 18, given that this has not been dealt with systematically throughout the revised manuscript – thus, obvious errors now appear in the Results section.

In the Introduction and Methods section it is concerning that the authors themselves seem not to have thought through the timeframe of their review fully enough to realise from the outset that studies published before 2020 could not, by definition, have dealt with the COVID-19 pandemic period which was only declared in 2020. Nonetheless, it is appreciated that they have now seen fit to re-consider their inclusion criteria and strip another three studies out of their analysis. However, rather than doing a careful reworking of their Abstract and Results and Discussion sections, their revision seems rather piecemeal and non-thorough. Their ongoing mentions of 2019 create confusion and need to be reconsidered, eg.

• Lines 145-146: “...studies that examined telework during the COVID-19 pandemic (2019 and onwards)” [my emphasis].

• Lines 155-156: “...studies that examined a move to telework arrangements before 2019”. [my emphasis – surely this should be “before the pandemic’s onset” or at least “before 2020”?]

• Lines 167-168: “We included studies published from January 2019 to December 2023 168 (inclusive) to fully capture the impacts of the COVID-19 pandemic”. [this timeframe needs to change – all pre-pandemic studies were excluded in the revised set]

• Last inclusion criterion in Table 1

• Last exclusion criterion in Table 1

In the PRISMA chart, the “wrong timeframe” needs to be corrected from n=22 to n=25 for all the numbers to make sense.

In the Results section there are various changes needed. The table of included studies needs its heading to be corrected (it still says ‘n=21’). It also does not follow the usual convention of those studies appearing consecutively in the reference list – instead, the reference numbers for the studies jump around in a confusing manner. It would be preferable to follow convention, although this will require the authors to put significant time into rejuggling their citations throughout the paper and finalising the reference list. Then, with regards to their statistical analyses, the authors need firstly to search for and remove all references to a data corpus of 21 studies and replace those with mentioning of 18 studies. Then they need to recheck their calculations in every respect and must make absolutely certain that none of their key findings disappeared or changed when the three studies were belatedly excluded.

Some examples of errors in the Results section include:

• Lines 256-257: it makes no sense to say “...with the other 19 (90.5%) having a single country focus [1,2,17,19,25,38–50]”. Please re-check exactly which studies should be cited here and recalculate the percentage out of 18. Likewise, please re-check the figures and citations in the rest of that paragraph (up to and including line 264).

• In the paragraph starting with line 265 the figures and especially percentages need to be re-checked and corrected. For instance, 16 out of 18 studies equates to 89%, not 76%, and the authors should double-check that all the studies cited were from their revised data corpus. Similarly, line 266 includes the nonsensical wording “...nine (56.3%) of those being cross-sectional [19, 38,40,42,44,45,47,48]” when only eight studies are cited; furthermore, nine out of 18 studies would equate to 50%, not 56.3%.

• In lines 271-272 there is also some immediately apparent incorrect information, where the authors say “Most of the studies (18; 86%) provided gender breakdowns...” – from a cursory check of the (revised) table of included studies, that wording needs to change to “All of the studies (18; 100%) provided gender breakdowns.” This then makes the sentence on lines 273-274 redundant so it can be removed altogether: “However, of the total participant set of 54,457 across all study samples, no gender was specified for 14,058 (26%)”.

• The authors need to search for and remove any references to the previous participant set of 54,457 – the revised data corpus contained a participant set of only 27,185.

• Line 254: “Majority of the studies were from European and North American continents (n = 18)” – since the revised dataset was 18 studies, the authors should check whether this should read “All the studies (18;100%)...”.

• Lines 270-273 are no longer correct.

• Line 281 – please recalculate the percentage: 17 out of 18 studies equates to 94%, not 81%. Also, please check that the correct studies have been cited.

• Lines 282-284 – again, the authors should check that the correct studies have been cited; they include ref [42] but that is one of the studies excluded in Table 2.

• Line 290 – “four (19%) of the studies...” is incorrect; it should read “four (22%) of the studies...”.

• Line 301 – ref [42], one of the excluded studies, is again cited.

• Line 303 –re-calculation is needed: four out of 18 studies equates to 22%, not 19%.

• Line 335 – “Ten studies found...that organizational antecedents can be attributed to high presenteeism 336 and absenteeism rates among teleworkers [1,17,19,37–38,40,44,48,49]” cites only 9 studies. Re-checking is needed, with addition of the relevant percentage.

• Lines 340-341 – here the authors report what seems to be quite a significant finding in that it was mentioned by six studies. Thus, while they have written “Due to high job demands, many workers felt compelled to continue working despite illness [1,17,37,40,43,44]”, this could perhaps be strengthened by changing the wording to something like this: “Six (33%) of the studies found that, due to high job demands, workers felt compelled to continue working despite illness [1, 17, 37, 40,43,44]”.

• Lines 360-363 – the authors say “Studies have also demonstrated that access to telework can impact incidence of 361 absenteeism [38]...” [reviewer’s emphasis]. As only one study found this it would be preferable to say “One study by Borge et al [38] assessed the relationship between telework access and sickness absenteeism and found that workers who lack the ability to access telework from home showed a higher likelihood of absenteeism”.

• Line 417 – the short sentence “Anxiety and worry related to online presenteeism [1]” can probably be removed as this is already covered in lines 410-412.

• Line 469 - Lines 487-488 – the authors should re-check the data and report it correctly; saying “We found that 19 studies included a definition of presenteeism” makes no sense, given that there were only 18 studies in the final analysis.

• Line 616 – “Lastly, there were only four out of 21 articles that reported a definition of absenteeism” needs to be fixed to reflect the revised data corpus of 18 articles and how many of the final included studies included a definition of absenteeism.

In the Discussion section there are various issues the authors need to consider to remove ambiguity and sharpen their argument.

• Lines 537-538 – “Potential gaps within the literature persist regarding absenteeism definitions as there were only four papers that explicitly defined absenteeism, which warrants further investigation” is a little unclear. Were those four papers included in the revised data corpus or could the authors only find four papers in the broader literature?

• Lines 554-555 – in the assertion “Similarities and differences persisted among the antecedents of presenteeism and absenteeism” it is unclear what is meant by “persisted”. Perhaps “were apparent in the review” would be more appropriate?

• Lines 577-578 – the wording “...could impact the environment of workers, which can ultimately impact employees’ decision...” makes it sound like there is a difference between “workers” and “employees”. The authors could keep things simple by saying something like “...could impact the work environment, which can ultimately impact employees’ decision...”.

• Lines 606-610 – in dealing with the limitations of their study, the authors must tweak their wording to match their final data corpus. Instead of saying “Firstly, the majority of the studies were conducted in High-Income Countries, primarily in Europe and North America” it would be better to say (if correct) “All of the studies were conducted in Europe and North America, regions which are dominated by High-Income Countries”. Likewise, “This geographical concentration suggests a potential bias towards perspectives and experiences from the Global North. Consequently, there is a significant gap in research from the Global South and, in particular, from Lower- and Middle-Income Countries” could be changed to something like “This geographical concentration suggests a potential bias towards perspectives and experiences from the Global North, with a dearth of research from the Global South where Low- and Middle-Income Countries are more prevalent”.

In relation to the Conclusion, this remains unusually long and the authors could certainly pare it down more. They could, for example, remove the following wording from lines 668-670: “i.e., job demands impacted an individual's work environment, which lead to individual adaptations of presenteeism and absenteeism behaviours”. Only their most significant, punchy findings should feature in the conclusion – the substantiating information for these having already appeared earlier in the manuscript.

Finally, here are a few basic grammatical, typographic etc errors which leapt out and need correcting – by no means an exhaustive list:

• Line 18 in abstract – “defined as” seems to be in a different font

• Line 44 – “15 -69” should read “15-69”

• Line 52 – clumsy wording, “...due to the COVID-19”. Rather revert to “...due to COVID-19”, or switch to “...due to the pandemic”.

• Line 87 “physically absent to work” is clumsy – preferable would be “physically absent from work” (for consistency with line 85). Also, there is a missing full stop after ref [28].

• A line needs to be added before the heading “2.5 Data Synthesis” (line 184).

• Table 2 – a blank line should be added between Grigore et al [25] and Hadi et al [41], for consistency and reader-friendliness; without this, at a glance it looks like only 17 studies are included.

• Also in Table 2, the authors should also settle on one style for the larger sample sizes – they use the format 4,329 for Borge et al, then 2530 for Okawara et al and 3532 for Takayama et al, and then 12 354 for Ryoo et al. A comma format (4,329) may be most reader-friendly, but all three styles are acceptable as long as there is consistency.

• Line 202 – a space is needed after [25].

• Line 304 – “understands” should read “understand”.

• Lines 315 and 390 – a space is needed after ref [25].

• Line 392 – “antecedents” should read “antecedent”

• Line 401 – the short sentence “Anxiety and worry related to online presenteeism [1]” can be removed as it repeats what appears in lines 394-396.

• Line 402 – “include” should read “including”

• The authors should fully act on the previous recommendation to consistently use the same tense when reporting findings from the included studies – some remain reported in past tense and others in present tense, which gets confusing for the reader. See for instance lines 412-413 “Brosi and Gerpott [38] noted...” versus line 417 “Walker et al [48] notes (sic)...”. Also, the paragraph starting with line 419 contains “Yildirim [36] further explores...” and “Adisa et al [1] also focused on...” [reviewer’s emphases].

• Line 419 – the dash after ‘presenteeism’ needs to be removed; it probably crept in when an unnecessary comma was deleted.

• Lines 427-429 – there is confusion regarding citations; “Mauricio and Laranjeira [25,42] and “Grigore [25,42]” that the authors need to correct.

• Line 434 – “Walker et al [50] notes...” should read “Walker et al [50] note...” or “...noted”.

• Also in relation to consistency, the authors should do a search on “covid” and replace each instance with either “COVID” or “COVID-19” across the whole manuscript, for consistency. An exception to this would be in the reference list and where studies which use different versions are directly quoted.

• Line 479 – insert a blank line before the heading “4.1 Definitions for Presenteeism and Absenteeism”.

• Table 3 – fifth line under sub-themes should read “Poor mental health”, not “Poor metal health”. Okawara et al is incorrectly referenced – [42] instead of [44]. The authors should also remove residual underlining which is not needed.

• Table 4 – a blank line should be added between themes, to be consistent with Table 3 and for greater reader-friendliness.

• Line 594 – “distinguishing” should read “distinguish”.

• Line 598 – “...environment in which one works in...” should be “...environment in which one works...”.

• Line 677 – “reduce” should read “reducing”.

• Reference list and supplementary materials – the authors need to re-check these carefully for typographical errors. For instance, in the tables of included and excluded studies some article titles and author names are in capital letters for no clear reason, there is inconsistent formatting – some two-author papers use “and” between the authors’ names whilst others use “&” and some studies have incomplete titles or inscrutable titles like “attendance” or “ROIDE RR.HH”. Wherever possible, rather than using “N/A” for authors it is also preferable (for credibility and trustworthiness) to name SOME sort of author (eg: an organisation) for each and every article sourced. On the Included Studies table, it is unclear why the word “dissertation” is in red for Michael 2021, and also why no date of data collection is shown – surely this should have been apparent on reading the dissertation?

7. PLOS authors have the option to publish the peer review history of their article (what does this mean? ). If published, this will include your full peer review and any attached files.

**Do you want your identity to be public for this peer review?** For information about this choice, including consent withdrawal, please see our Privacy Policy .

Reviewer #1: No

Reviewer #2: No

---

## [Decision Letter · Decision Letter 3]

29 Jan 2025

PMEN-D-24-00187R3

Sick leave or Work Sick? Examining the Antecedents and Conceptualizations of Presenteeism and Absenteeism among Teleworkers During COVID-19: A Scoping Review

PLOS Mental Health

Dear Dr. Nowrouzi-Kia,

Thank you for submitting your manuscript to PLOS Mental Health. After careful consideration, we feel that it has merit but does not fully meet PLOS Mental Health’s publication criteria as it currently stands. Therefore, we invite you to submit a revised version of the manuscript that addresses the points raised during the review process.

We look forward to receiving your revised manuscript.

Kind regards,

Bochra Nourhene Saguem, M.D.

Academic Editor

PLOS Mental Health

Journal Requirements:

Additional Editor Comments (if provided):

Reviewers' comments:

Reviewer's Responses to Questions

**Comments to the Author**

1. If the authors have adequately addressed your comments raised in a previous round of review and you feel that this manuscript is now acceptable for publication, you may indicate that here to bypass the “Comments to the Author” section, enter your conflict of interest statement in the “Confidential to Editor” section, and submit your "Accept" recommendation.

Reviewer #3: All comments have been addressed

Reviewer #4: (No Response)

Reviewer #5: (No Response)

2. Does this manuscript meet PLOS Mental Health’s publication criteria ? Is the manuscript technically sound, and do the data support the conclusions? The manuscript must describe methodologically and ethically rigorous research with conclusions that are appropriately drawn based on the data presented.

Reviewer #3: Yes

Reviewer #4: Partly

Reviewer #5: Yes

3. Has the statistical analysis been performed appropriately and rigorously?

Reviewer #3: Yes

Reviewer #4: N/A

Reviewer #5: Yes

4. Have the authors made all data underlying the findings in their manuscript fully available (please refer to the Data Availability Statement at the start of the manuscript PDF file)?

Reviewer #3: Yes

Reviewer #4: Yes

Reviewer #5: Yes

5. Is the manuscript presented in an intelligible fashion and written in standard English?

Reviewer #3: (No Response)

Reviewer #4: Yes

Reviewer #5: Yes

6. Review Comments to the Author

Reviewer #3: Attached

Reviewer #4: PubMed is commonly used for scoping reviews due to its broader scope, while MEDLINE, often accessed via platforms like Ovid, is typically preferred for systematic reviews due to its curated and rigorous indexed content

Other databases could be explored, such as EBSCO, LitCovid (subset of PubMed)/COVID-19 Research Database, Google Scholar/ILO.

Other key search terms/words could be work from home, workplace productivity, employee health, mental health, burnout , stress, flexible work.

Absenteeism and Presenteeism reasons, Outcomes/Findings/Conclusions could be added to the summary table.

Line 182-183: The software version is to be stated.

Line 228-229: The sentence is unclear and requires revision.

Line 236: The reason is to be provided.

There were two figures 1, one on Page 12, Figure 1, and the other as an Attachment. Page 12 Figure is to be deleted as it is the uncorrected version.

At least one decimal point for the percentage figure is to be presented where applicable in the text and table(s).

Line 306: Cap ‘T ’for table 4

Table 4 COVID-related Stressors (n=1): There were two references cited. Is it n=1 or n=2? [25] missing for Grigore. ‘n=2’ for ‘Poor Mental Health (2)’

The list of references did not conform to the journal format.

Reviewer #5: Thanks for a clear article.

Earlier you state that the work was done to select the studies to

November but line 174 states some papers could be published in December. Was this possible?

Line 399- is the end of the sentence missing?

7. PLOS authors have the option to publish the peer review history of their article (what does this mean? ). If published, this will include your full peer review and any attached files.

**Do you want your identity to be public for this peer review?** For information about this choice, including consent withdrawal, please see our Privacy Policy .

Reviewer #3: No

Reviewer #4: No

Reviewer #5: No

---

## [Decision Letter · Decision Letter 4]

13 Mar 2025

Sick leave or Work Sick? Examining the Antecedents and Conceptualizations of Presenteeism and Absenteeism among Teleworkers During COVID-19: A Scoping Review

PMEN-D-24-00187R4

Dear Dr. Nowrouzi-Kia,

We are pleased to inform you that your manuscript 'Sick leave or Work Sick? Examining the Antecedents and Conceptualizations of Presenteeism and Absenteeism among Teleworkers During COVID-19: A Scoping Review' has been provisionally accepted for publication in PLOS Mental Health.

Best regards,

Bochra Nourhene Saguem, M.D.

Academic Editor

PLOS Mental Health

Reviewer Comments (if any, and for reference):

Reviewer's Responses to Questions

**Comments to the Author**

1. If the authors have adequately addressed your comments raised in a previous round of review and you feel that this manuscript is now acceptable for publication, you may indicate that here to bypass the “Comments to the Author” section, enter your conflict of interest statement in the “Confidential to Editor” section, and submit your "Accept" recommendation.

Reviewer #3: All comments have been addressed

Reviewer #4: (No Response)

Reviewer #5: All comments have been addressed

2. Does this manuscript meet PLOS Mental Health’s publication criteria ? Is the manuscript technically sound, and do the data support the conclusions? The manuscript must describe methodologically and ethically rigorous research with conclusions that are appropriately drawn based on the data presented.

Reviewer #3: Yes

Reviewer #4: Partly

Reviewer #5: (No Response)

3. Has the statistical analysis been performed appropriately and rigorously?

Reviewer #3: Yes

Reviewer #4: N/A

Reviewer #5: (No Response)

4. Have the authors made all data underlying the findings in their manuscript fully available (please refer to the Data Availability Statement at the start of the manuscript PDF file)?

Reviewer #3: Yes

Reviewer #4: Yes

Reviewer #5: (No Response)

5. Is the manuscript presented in an intelligible fashion and written in standard English?

Reviewer #3: Yes

Reviewer #4: Yes

Reviewer #5: (No Response)

6. Review Comments to the Author

Reviewer #3: The article is well presented and should be considered for publication.

Reviewer #4: The authors have addressed most of the comments.

Reviewer #5: (No Response)

7. PLOS authors have the option to publish the peer review history of their article (what does this mean? ). If published, this will include your full peer review and any attached files.

**Do you want your identity to be public for this peer review?** For information about this choice, including consent withdrawal, please see our Privacy Policy .

Reviewer #3: No

Reviewer #4: No

Reviewer #5: No
